# A Unified Framework for Rank-based Loss Minimization

**Rufeng Xiao**[*]     **Yuze Ge**[*]     **Rujun Jiang**[†]     **Yifan Yan**
School of Data Science, Fudan University
{rfxiao21,yzge23}@m.fudan.edu.cn
rjjiang@fudan.edu.cn
yanyf21@m.fudan.edu.cn

## Abstract

The empirical loss, commonly referred to as the average loss, is extensively utilized for training machine learning models. However, in order to address the diverse performance requirements of machine learning models, the use of the rank-based loss is prevalent, replacing the empirical loss in many cases. The rank-based loss comprises a weighted sum of sorted individual losses, encompassing both convex losses like the spectral risk, which includes the empirical risk and conditional value-at-risk, and nonconvex losses such as the human-aligned risk and the sum of the ranked range loss. In this paper, we introduce a unified framework for the optimization of the rank-based loss through the utilization of a proximal alternating direction method of multipliers. We demonstrate the convergence and convergence rate of the proposed algorithm under mild conditions. Experiments conducted on synthetic and real datasets illustrate the effectiveness and efficiency of the proposed algorithm.

## 1 Introduction

The empirical risk function is the cornerstone of machine learning. By minimizing this function, machine learning models typically achieve commendable average performance. However, as these models find applications across a diverse spectrum of real-world scenarios, the evaluation standards for machine learning performance have evolved to include factors such as risk aversion and fairness [49, 9]. This has led to the empirical loss function often being supplanted by other loss functions, many of which fall under the category of the rank-based loss, which consists of a weighted sum of sorted individual losses. We consider the following optimization problem in this paper, which minimizes the rank-based loss plus a regularizer:

$$\min_{\boldsymbol{w} \in \mathbb{R}^d} \quad \sum_{i=1}^{n} \sigma_i \boldsymbol{l}_{[i]} \left( -\boldsymbol{y} \odot (X\boldsymbol{w}) \right) + g(\boldsymbol{w}). \tag{1}$$

Here, $\boldsymbol{l}(\boldsymbol{u}) \triangleq [l(u_1), \ldots, l(u_n)]^\top : \mathbb{R}^n \to \mathbb{R}^n$ represents a vector-valued mapping where $l : \mathbb{R} \to \mathbb{R}$ denotes an individual loss function and $u_i$ is the $i$-th element of $\boldsymbol{u}$, $\boldsymbol{l}_{[1]}(\cdot) \leqslant \cdots \leqslant \boldsymbol{l}_{[n]}(\cdot)$ denotes the order statistics of the empirical loss distribution, $g : \mathbb{R}^d \to \mathbb{R}$ is a regularizer that induces desired structures, $\boldsymbol{w} \in \mathbb{R}^d$ represents the parameters of the linear model, $X \in \mathbb{R}^{n \times d}$ is the data matrix, and $\boldsymbol{y} \in \{\pm 1\}^n$ is the label vector. The subscript $[i]$ denotes the $i$-th smallest element, $\sigma_i \in [0, 1]$ is the weight corresponding to the $i$-th smallest element, and "$\odot$" represents the Hadamard product. Depending on the values of $\sigma_i$, the rank-based loss can encompass a significant range of losses often

---

[*]Equal contribution
[†]Corresponding author

37th Conference on Neural Information Processing Systems (NeurIPS 2023).

used in machine learning and other fields: the L-risk or the spectral risk [1, 2, 35], the trimmed risk or the sum of the ranked-range loss [27], the human-aligned risk based on the cumulative prospect theory [31].

**Related work.** When the values of $\sigma_i$ are constant and monotonically increase, i.e., $\sigma_1 \leq \cdots \leq \sigma_n$, the rank-based loss is referred to as the spectral risk [1]. Several methods have been proposed in the literature, including: a derivative-free method proposed by [25] that utilizes a stochastic smoothing technique; an adaptive sampling algorithm proposed by [14] for the minimization of conditional value at risk (CVaR), which is a special case of the spectral risk; and a stochastic algorithm developed by [36] for minimizing the general spectral risk by characterizing the subdifferential of the rank-based loss. In the case of nonconvex rank-based losses, [27] utilizes the difference of convex algorithm for minimizing the average of ranked-range (AoRR) loss. The nonconvex human-aligned risk was minimized by [31] by directly computing the gradient of the function, even though the rank-based loss is nondifferentiable, without any convergence guarantee. Despite the increasing applications of the rank-based loss, a unified framework for addressing the rank-based loss minimization remains elusive. Moreover, stochastic methods may encounter bias issues [36], existing methods for the spectral risk cannot deal with nonconvex rank-based losses, and existing methods for human-aligned risk lack convergence guarantee [31].

**Our Contributions.** We present a unified framework for the minimization of the rank-based loss to overcome the challenges of existing methods. Specifically, we focus on monotone increasing loss functions, such as the hinge loss and the logistic loss, and use weakly convex regularizers. We leverage the alternating direction multiplier method (ADMM), which is widely used in solving nonsmooth nonconvex composite problems [8, 32, 48]. To utilize the ADMM framework, we introduce an auxiliary variable to represent the product of the data matrix and parameter vector. The two subproblems are either strongly convex (with a proximal term), which can be solved by various existing methods, or can be efficiently solved by the pool adjacent violators algorithm (PAVA) [6]. We demonstrate that our algorithm can find an $\epsilon$-KKT point in at most $O(1/\epsilon^2)$ iterations under mild conditions. To relax the assumptions further, we propose a variant of the ADMM with a smoothing technique when the regularizer is nonsmooth. Notably, when the Moreau envelope is applied to smooth the nonsmooth regularizer, we show that our algorithm can find an $\epsilon$-KKT point within $O(1/\epsilon^4)$ iterations.

Our contributions can be summarized as follows:

1) We propose a unified framework for the rank-based loss minimization for monotonically increasing loss functions. This approach is versatile, effectively dealing with both convex and nonconvex loss functions by allowing different settings of weights $\sigma_i$.
2) Furthermore, the regularizer in our problem can be weakly convex functions, extending existing works that only consider convex or smooth regularizers, such as the $l_1$ and $l_2$ penalty.
3) We theoretically demonstrate that our ADMM algorithm converges to an $\epsilon$-approximate KKT point under different assumptions.
4) The experiments in three different aggregate loss function frameworks demonstrate the advantages of the proposed algorithm.

## 2 Preliminaries

In this section, we explore the wide variety of rank-based losses and introduce the Moreau envelope as a smooth approximation of weakly convex functions.

### 2.1 Rank-Based Loss

Let $\{(X_1, y_1), \ldots, (X_n, y_n)\}$ be an i.i.d. sample set from a distribution $\mathbb{P}$ over a sample space $(\mathcal{X}, \mathcal{Y})$, and $\mathcal{L}$ be the individual loss function. Then $Y = \mathcal{L}(\boldsymbol{w}, X, y)$ is also a random variable, and $Y_i = \mathcal{L}(\boldsymbol{w}, X_i, y_i)$ represents the $i$-th training loss with respect to the sample $(X_i, y_i)$, $i \in \{1, \ldots, n\}$. Let $Y_{[1]} \leq \ldots \leq Y_{[n]}$ be the order statistics. The rank-based loss function in problem (1) equals

$$\sum_{i=1}^{n} \sigma_i Y_{[i]}. \tag{2}$$

**Spectral Risk**  The spectral risk, as defined in [1], is given by:

$$R_\sigma(\boldsymbol{w}) = \int_0^1 \text{VaR}_t(\boldsymbol{w})\sigma(t)\,dt, \tag{3}$$

where $\text{VaR}_t(\boldsymbol{w}) = \inf\{u : F_{\boldsymbol{w}}(u) \geq t\}$ denotes the quantile function, and $F_{\boldsymbol{w}}$ represents the cumulative distribution function of $Y$. The function $\sigma : [0,1] \to \mathbb{R}_+$ is a nonnegative, nondecreasing function that integrates to 1, known as the spectrum of the rank-based loss. The discrete form of (3) is consistent with (2), with $\sigma_i = \int_{(i-1)/n}^{i/n} \sigma(t)\,dt$ [36]. The spectral risk builds upon previous aggregate losses that have been widely used to formulate learning objectives, such as the average risk [46], the maximum risk [43], the average top-$k$ risk [21] and conditional value-at-risk [39]. By choosing a different spectral $\sigma(t)$, such as the superquantile [30], the extremile [15], and the exponential spectral risk measure [11], can be constructed for various spectral risks.

**Human-Aligned Risk**  With the growing societal deployment of machine learning models for aiding human decision-making, these models must possess qualities such as fairness, in addition to strong average performance. Given the significance of these additional requirements, amendments to risk measures and extra constraints have been introduced [28, 23, 19]. Inspired by the cumulative prospect theory (CPT) [45], which boasts substantial evidence for its proficiency in modeling human decisions, investigations have been carried out in bandit [24] and reinforcement learning [41]. Recently, [31] also utilized the concept of CPT to present a novel notion of the empirical human risk minimization (EHRM) in supervised learning. The weight assigned in (3) is defined as $\sigma_i = w_{a,b}(\frac{i}{n})$, where

$$w_{a,b}(t) = \frac{3 - 3b}{a^2 - a + 1}\left(3t^2 - 2(a+1)t + a\right) + 1.$$

These weights assign greater significance to extreme individual losses, yielding an S-shaped CPT-weighted cumulative distribution. Moreover, we consider another CPT-weight commonly employed in decision making [45, 41]. Let $\boldsymbol{z} = -\boldsymbol{y} \odot (X\boldsymbol{w})$ and $B$ be a reference point. Unlike previous weight settings, $\sigma_i$ is related to the value of $z_{[i]}$ and is defined as

$$\sigma_i(z_{[i]}) = \begin{cases} \omega_-\big(\frac{i}{n}\big) - \omega_-\big(\frac{i-1}{n}\big), & z_{[i]} \leq B, \\ \omega_+\big(\frac{n-i+1}{n}\big) - \omega_+\big(\frac{n-i}{n}\big), & z_{[i]} > B, \end{cases} \tag{4}$$

where

$$\omega_+(p) = \frac{p^\gamma}{(p^\gamma + (1-p)^\gamma)^{1/\gamma}}, \quad \omega_-(p) = \frac{p^\delta}{(p^\delta + (1-p)^\delta)^{1/\delta}}, \tag{5}$$

with $\gamma$ and $\delta$ as hyperparameters.

**Ranked-Range Loss**  The average of ranked-range aggregate (AoRR) loss follows the same structure as (2), where

$$\sigma_i = \begin{cases} \frac{1}{k-m}, & i \in \{m+1, \ldots, k\}, \\ 0, & \text{otherwise}, \end{cases} \tag{6}$$

with $1 \leq m < k \leq n$. The ranked-range loss effectively handles outliers, ensuring the robustness of the model against anomalous observations in the dataset [27]. It is clear that AoRR includes the average loss, the maximum loss, the average top-$k$ loss, value-at-risk [39] and the median loss [34]. [27] utilized the difference-of-convex algorithm (DCA) [40] to solve the AoRR aggregate loss minimization problem, which can be expressed as the difference of two convex problems.

## 2.2  Weakly Convex Function and Moreau Envelope

A function $g : \mathbb{R}^d \to \mathbb{R}$ is said to be $c$-weakly convex for some $c > 0$ if the function $g + \frac{c}{2}\|\cdot\|^2$ is convex. The class of weakly convex functions is extensive, encompassing all convex functions as well as smooth functions with Lipschitz-continuous gradients [37]. In our framework, we consider weakly convex functions as regularizers. It is worth noting that weakly convex functions constitute a rich class of regularizers. Convex $L^p$ norms with $p \geq 1$ and nonconvex penalties such as the Minimax Concave Penalty (MCP) [52] and the Smoothly Clipped Absolute Deviation (SCAD) [20] are examples of weakly convex functions [7].

Next, we define the Moreau envelope of $c$-weakly convex function $g(\boldsymbol{w})$, with proximal parameter $0 < \gamma < \frac{1}{c}$:

$$M_{g,\gamma}(\boldsymbol{w}) = \min_{\boldsymbol{x}} \left\{ g(\boldsymbol{x}) + \frac{1}{2\gamma} \|\boldsymbol{x} - \boldsymbol{w}\|^2 \right\}. \tag{7}$$

The proximal operator of $g$ with parameter $\gamma$ is given by

$$\mathrm{prox}_{g,\gamma}(\boldsymbol{w}) = \arg\min_{\boldsymbol{x}} \left\{ g(\boldsymbol{x}) + \frac{1}{2\gamma} \|\boldsymbol{x} - \boldsymbol{w}\|^2 \right\}.$$

We emphasize that $\mathrm{prox}_{g,\gamma}(\cdot)$ is a single-valued mapping, and $M_{g,\gamma}(\boldsymbol{w})$ is well-defined since the objective function in (7) is strongly convex for $\gamma \in \left(0, c^{-1}\right)$ [7].

The Moreau envelope is commonly employed to smooth weakly convex functions. From [7, Lemma 3.1-3.2] we have

$$\nabla M_{g,\gamma}(\boldsymbol{w}) = \gamma^{-1} \left( \boldsymbol{w} - \mathrm{prox}_{g,\gamma}(\boldsymbol{w}) \right) \in \partial g \left( \mathrm{prox}_{g,\gamma}(\boldsymbol{w}) \right). \tag{8}$$

Here $\partial$ represents Clarke generalized gradient [10]. For locally Lipschitz continuous function $f : \mathbb{R}^d \to \mathbb{R}$, its Clarke generalized gradient is denoted by $\partial f(\boldsymbol{x})$. We say $\boldsymbol{x}$ is stationary for $f$ if $\boldsymbol{0} \in \partial f(\boldsymbol{x})$. For convex functions, Clarke generalized gradient coincides with subgradient in the sense of convex analysis. Under assumptions in Section 3.3, both $\Omega(\boldsymbol{z}) := \sum_{i=1}^{n} \sigma_i \boldsymbol{l}_{[i]}(\boldsymbol{z})$ and $g(\boldsymbol{w})$ are locally Lipschitz continuous [12, 47]. Thus, the Clarke generalized gradients of $\Omega(\boldsymbol{z})$ and $g(\boldsymbol{w})$ are well defined, and so is that of $L_\rho(\boldsymbol{w}, \boldsymbol{z}; \boldsymbol{\lambda})$ defined in Section 3.1.

## 3 Optimization Algorithm

Firstly, we make the basic assumption about the loss function $l$ and the regularizer $g$:

**Assumption 1** *The individual loss function $l(\cdot) : \mathcal{D} \to \mathbb{R}$ is convex and monotonically increasing, where $\mathcal{D} \subset \mathbb{R}$ is the domain of $l$. The regularizer $g(\cdot) : \mathbb{R}^d \to \mathbb{R} \cup \{\infty\}$ is proper, lower semicontinuous, and $c$-weakly convex. Furthermore, $l(\cdot)$ and $g(\cdot)$ are lower bounded by 0.*

Many interesting and widely used individual loss functions satisfy this assumption, such as logistic loss, hinge loss, and exponential loss. Our algorithm described below can also handle monotonically decreasing loss functions, by simply replacing $\sigma_i$ with $\sigma_{n-i+1}$ in (1). We assume $l$ and $g$ are lower bounded, and take the infimum to be 0 for convenience. The lower bounded property of $l$ implies that the $\Omega$ is also lower bounded by 0 since $\sigma_i \geq 0$ for all $i$.

### 3.1 The ADMM Framework

By Assumption 1, the original problem is equivalent to

$$\min_{\boldsymbol{w}} \sum_{i=1}^{n} \sigma_i l \left( (D\boldsymbol{w})_{[i]} \right) + g(\boldsymbol{w}),$$

where $l$ is the individual loss function, $D = -\mathrm{diag}(\boldsymbol{y})X$ and $\mathrm{diag}(\boldsymbol{y})$ denotes the diagonal matrix whose diagonal entries are the vector $\boldsymbol{y}$.

To make use of the philosophy of ADMM, by introducing an auxiliary variable $\boldsymbol{z} = D\boldsymbol{w}$, we reformulate the problem as

$$\min_{\boldsymbol{w}, \boldsymbol{z}} \quad \Omega(\boldsymbol{z}) + g(\boldsymbol{w}) \qquad \text{s.t.} \quad \boldsymbol{z} = D\boldsymbol{w}, \tag{9}$$

where $\Omega(\boldsymbol{z}) = \sum_{i=1}^{n} \sigma_i l \left( z_{[i]} \right)$.

Note that for human-aligned risk, $\sigma_i$ should be written as $\sigma_i(z_{[i]})$ since it is a piecewise function with two different values in (4), but for simplicity, we still write it as $\sigma_i$.

The augmented Lagrangian function can then be expressed as

$$L_\rho(\boldsymbol{w}, \boldsymbol{z}; \boldsymbol{\lambda}) = \sum_{i=1}^{n} \Omega(\boldsymbol{z}) + \frac{\rho}{2} \|\boldsymbol{z} - D\boldsymbol{w} + \frac{\boldsymbol{\lambda}}{\rho}\|^2 + g(\boldsymbol{w}) - \frac{\|\boldsymbol{\lambda}\|^2}{2\rho}.$$

The ADMM, summarised in Algorithm 1, cyclically updates $w, z$ and $\lambda$, by solving the $w$- and $z$-subproblems and adopting a dual ascent step for $\lambda$. In Algorithm 1, we assume the $w$- subproblem can be solved exactly (for simplicity). When $g(w)$ is a smooth function, the $w$-subproblem is a smooth problem and can be solved by various gradient-based algorithms. Particularly, the $w$-subproblem is the least squares and admits a closed-form solution if $g(w) \equiv 0$ or $g(w) = \frac{\mu}{2}||w||_2^2$, where $\mu > 0$ is a regularization parameter. When $g(w)$ is nonsmooth, if $\text{prox}_{g,\gamma}(w)$ is easy to compute, we can adopt the proximal gradient method or its accelerated version [4]. Particularly, if $g(w) = \frac{\mu}{2}||w||_1$, then the subproblem of $w$ becomes a LASSO problem, solvable by numerous effective methods such as the Coordinate Gradient Descent Algorithm (CGDA) [22], the Smooth L1 Algorithm (SLA) [42], and the Fast Iterative Shrinkage-Thresholding Algorithms (FISTA) [4].

---

**Algorithm 1** ADMM framework

**Input:** $X$, $y$, $w^0$, $z^0$, $\lambda^0$, $\rho$, $r$, and $\sigma_i, i = 1, ..., n$.

1: **for all** $k = 0, 1, ...$ **do**
2: $\quad z^{k+1} = \arg\min_z \Omega(z) + \frac{\rho}{2}||z - Dw^k + \frac{\lambda^k}{\rho}||^2$.
3: $\quad w^{k+1} = \arg\min_w \frac{\rho}{2}||z^{k+1} - Dw + \frac{\lambda^k}{\rho}||^2 + g(w) + \frac{r}{2}||w - w^k||^2$.
4: $\quad \lambda^{k+1} = \lambda^k + \rho(z^{k+1} - Dw^{k+1})$.
5: **end for**

---

At first glance, the $z$-subproblem is nonconvex and difficult to solve. However, we can solve an equivalent convex chain-constrained program that relies on sorting and utilize the pool adjacent violators algorithm (PAVA). More specifically, we follow the approach presented in [13, Lemma 3] and introduce the auxiliary variable $m = Dw^k - \frac{\lambda^k}{\rho}$ (we remove the superscript for $m$ for simplicity). This enables us to express the $z$-subproblem in the equivalent form below

$$\min_z \quad \sum_{i=1}^n \sigma_i l(z_{p_i}) + \frac{\rho}{2}(z_{p_i} - m_{p_i})^2 \qquad \text{s.t. } z_{p_1} \leq z_{p_2} \leq \cdots \leq z_{p_n},$$

where $\{p_1, p_2, \ldots, p_n\}$ is a permutation of $\{1, \ldots, n\}$ such that $m_{p_1} \leqslant m_{p_2} \leqslant \cdots \leqslant m_{p_n}$.

## 3.2 The Pool Adjacent Violators Algorithm (PAVA)

To introduce our PAVA, for ease of notation and without loss of generality, we consider $m_1 \leqslant m_2 \leqslant \cdots \leqslant m_n$, i.e., the following problem

$$\min_z \quad \sum_{i=1}^n \sigma_i l(z_i) + \frac{\rho}{2}(z_i - m_i)^2 \qquad \text{s.t. } z_1 \leq z_2 \leq \cdots \leq z_n. \tag{10}$$

The above problem constitutes a convex chain-constrained program. Although it can be solved by existing convex solvers, the PAVA [6] is often more efficient.

The PAVA is designed to solve the following problem

$$\min_z \quad \sum_{i=1}^n \theta_i(z_i) \qquad \text{s.t. } z_1 \leq z_2 \leq \cdots \leq z_n,$$

where each $\theta_i$ represents a univariate *convex* function. In our problem, $\theta_i(z_i) = \sigma_i l(z_i) + \frac{\rho}{2}(z_i - m_i)^2$. The PAVA maintains a set $J$ that partitions the indices $\{1, 2, \ldots, n\}$ into consecutive blocks $\{[s_1 + 1, s_2], [s_2 + 1, s_3], \cdots, [s_k + 1, s_{k+1}]\}$ with $s_1 = 0$ and $s_{k+1} = N$. Here, $[a, b]$ represents the index set $\{a, a + 1, \ldots, b\}$ for positive integers $a < b$, and by convention, we define $[a, a] = a$. A block $[p, q]$ is termed a *single-valued block* if every $z_i$ in the optimal solution of the following problem has the same value,

$$\min_{z_p, \ldots, z_q} \quad \sum_{i=p}^q \theta_i(z_i) \qquad \text{s.t. } z_p \leq z_{p+1} \leq \cdots \leq z_q,$$

i.e., $z_p^* = z_{p+1}^* = \cdots = z_q^*$. In line with existing literature, we use $v_{[p,q]}$ to denote this value. For two consecutive blocks $[p, q]$ and $[q + 1, r]$, if $v_{[p,q]} \leq v_{[q+1,r]}$, the

two blocks are *in-order*; otherwise, they are *out-of-order*. Similarly, the consecutive blocks $\{[s_k, s_{k+1}], [s_{k+1} + 1, s_{k+2}], \cdots, [s_{k+t} + 1, s_{k+t+1}]\}$ are said to be *in-order* if $v_{[s_k s_{k+1}]} \leq v_{[s_{k+1}+1, s_{k+2}]} \leq \cdots \leq v_{[s_{k+t}+1, s_{k+t+1}]}$; otherwise they are *out-of-order*. Particularly, if $v_{[s_k s_{k+1}]} > v_{[s_{k+1}+1, s_{k+2}]} > \cdots > v_{[s_{k+t}+1, s_{k+t+1}]}$, the consecutive blocks are said to be *consecutive out-of-order*. The PAVA initially partitions each integer from 1 to $n$ into single-valued blocks $[i, i]$ for $i = 1, \ldots, n$. When there are consecutive *out-of-order* single-valued blocks $[p, q]$ and $[q + 1, r]$, the PAVA merges these two blocks by replacing them with the larger block $[p, r]$. The PAVA terminates once all the single-valued blocks are in-order.

---

**Algorithm 2** A refined pool-adjacent-violators algorithm for solving problem (10)

---

**Input:** $J = \{[1, 1], [2, 2], \ldots, [n, n]\}, \theta_i, i = 1, 2, \ldots, n.$
**Output:** $\{y_1^*, y_2^*, \ldots, y_N^*\}$
 1: **for** each $[i, i] \in J$ **do**
 2:     Compute the minimizer $v_{[i,i]}$ of $\theta_i(\boldsymbol{y})$
 3: **end for**
 4: **while** exists *out-of-order* blocks in $J$ **do**
 5:     Find consecutive *out-of-order* blocks $C = \{[s_k, s_{k+1}], [s_{k+1} + 1, s_{k+2}], \cdots, [s_{k+t} + 1, s_{k+t+1}]\}$.
 6:     $J \leftarrow J \backslash C \cup \{[s_k, s_{k+t+1}]\}$, compute the minimizer $v_{[s_k, s_{k+t+1}]}$ of $\sum_{i=s_k}^{s_{k+t+1}} \theta_i(\boldsymbol{z})$.
 7: **end while**
 8: **for** each $[m, n] \in J$ **do**
 9:     $y_i^* = v_{[m,n]}, \forall i = m, m + 1, \ldots, n.$
10: **end for**

---

Traditional PAVA processes the merging of *out-of-order* blocks one by one, identifying two consecutive *out-of-order* blocks and then solving a single unconstrained convex minimization problem to merge them. For constant $\sigma_i$, our proposed Algorithm 2, however, leverages the unique structure of our problem to improve the PAVA's efficiency. Specifically, we can identify and merge multiple consecutive *out-of-order* blocks, thereby accelerating computation. This acceleration is facilitated by the inherent properties of our problem. Notably, in case the spectral risk and the rank-ranged loss, the function $\theta_i(z_i)$ demonstrates strong convexity due to the presence of the quadratic term $\frac{\rho}{2}(z_i - m_i)^2$ and the convex nature of $l(z_i)$. This key observation leads us to the following proposition.

**Proposition 1** *For constant $\sigma_i$, suppose $v_{[m,n]} > v_{[n+1,p]}$, the blocks $[m, n]$ and $[n + 1, p]$ are consecutive out-of-order blocks. We merge these two blocks into $[m, p]$. Then the block optimal value, denoted by $v_{[m,p]}$, satisfies $v_{[n+1,p]} \leq v_{[m,p]} \leq v_{[m,n]}$.*

Proposition 1 provides a crucial insight: when we encounter consecutive *out-of-order* blocks with a length greater than 2, we can merge them simultaneously, rather than performing individual block merges. This approach significantly improves the efficiency of the algorithm. To illustrate this concept, consider a scenario where we have a sequence of values arranged as $v_{[1,2]} > v_{[3,4]} > v_{[5,6]}$. Initially, we merge the blocks $[1, 2]$ and $[3, 4]$, resulting in $v_{[1,4]} \geq v_{[3,4]} > v_{[5,6]}$. However, since $v_{[1,4]}$ is still greater than $v_{[5,6]}$, we need to merge the blocks $[1, 4]$ and $[5, 6]$ as well. Rather than calculating $v_{[1,4]}$ separately, we can streamline the process by directly merging the entire sequence into a single block, namely $[1, 6]$, in a single operation. By leveraging this approach, we eliminate the need for intermediate calculations and reduce the computational burden associated with merging individual blocks. This results in a more efficient version of the PAVA. Further details can be found in Appendix B.3.

It is worth noting that when $\theta_i(z_i)$ is convex with respect to $z_i$, the PAVA guarantees to find a global minimizer [6]. However, when considering the empirical human risk minimization with CPT-weight in (4), the function $\theta_i(z_i)$ is nonconvex, which is because $\sigma_i$ is a piecewise function with two different values. In such cases, we can still find a point that satisfies the first-order condition [12, Theorem 3]. In summary, we always have

$$\boldsymbol{0} \in \partial_{\boldsymbol{z}} L_\rho(\boldsymbol{w}^{k+1}, \boldsymbol{z}; \boldsymbol{\lambda}^k). \tag{11}$$

## 3.3 Convergence Analysis of Algorithm 1

We now demonstrate that Algorithm 1 is guaranteed to converge to an $\epsilon$-KKT point of problem (9). To this end, we shall make the following assumptions.

**Assumption 2** *The sequence* $\{\boldsymbol{\lambda}^k\}$ *is bounded and satisfies* $\sum_{k=1}^{\infty} \|\boldsymbol{\lambda}^k - \boldsymbol{\lambda}^{k+1}\|^2 < \infty$.

It is worth noting that Assumption 2 is commonly employed in ADMM approaches [51, 44, 3].

Next, we present our convergence result based on the aforementioned assumptions. As mentioned in Section 3.2, the PAVA can always find a solution for the $\boldsymbol{z}$-subproblem that satisfies the first-order condition (11). When $\sigma_i$ is constant, which is the case of the spectral risk and the AoRR risk, the subproblems of PAVA are strongly convex, and we can observe the descent property of the $\boldsymbol{z}$-subproblem:

$$L_\rho(\boldsymbol{z}^k, \boldsymbol{w}^k, \boldsymbol{\lambda}^k) - L_\rho(\boldsymbol{z}^{k+1}, \boldsymbol{w}^k, \boldsymbol{\lambda}^k) \geq 0. \tag{12}$$

However, if $\sigma_i$ has different values for $\boldsymbol{z}$ less than and larger than the reference point $B$ as in (4), the subproblems of PAVA may lose convexity, and the descent property may not hold. Thus, we assume that (12) holds for simplicity. By noting that the $w$-subproblem can be solved exactly as it is strongly convex, we make the following assumption.

**Assumption 3** *The* $\boldsymbol{w}$*-subproblem is solved exactly, and the* $\boldsymbol{z}$*-subproblem is solved such that* (11) *and* (12) *hold.*

**Theorem 1** *Assume that Assumptions 1, 2 and 3 hold. Then Algorithm 1 can find an $\epsilon$-KKT point* $(\boldsymbol{z}^{k+1}, \boldsymbol{w}^{k+1}, \boldsymbol{\lambda}^{k+1})$ *of problem* (9) *within* $O(1/\epsilon^2)$ *iterations, i.e.,*

$$\text{dist}\left(-\boldsymbol{\lambda}^{k+1}, \partial\Omega\left(\boldsymbol{z}^{k+1}\right)\right) \leq \epsilon, \quad \text{dist}\left(D^\top \boldsymbol{\lambda}^{k+1}, \partial g\left(\boldsymbol{w}^{k+1}\right)\right) \leq \epsilon, \quad \|\boldsymbol{z}^{k+1} - D\boldsymbol{w}^{k+1}\| \leq \epsilon,$$

*where* $\text{dist}(\boldsymbol{a}, A) = \min\{\|\boldsymbol{a} - \boldsymbol{x}\| : \boldsymbol{x} \in A\}$ *defines the distance of a point* $\boldsymbol{a}$ *to a set* $A$.

In the absence of Assumption 2, a common assumption for nonconvex ADMM is that $g(\boldsymbol{w})$ possesses a Lipschitz continuous gradient [48, 32], e.g., $g(\boldsymbol{w}) = \frac{\mu}{2}\|\boldsymbol{w}\|^2$. Under this assumption, Algorithm 1 guarantees to find an $\epsilon$-KKT point within $O(1/\epsilon^2)$ iterations [48, 32].

## 4 ADMM for a Smoothed Version of Problem (1)

In Section 3.3, our convergence result is established under Assumption 2 when a nonsmooth regularizer $g(\boldsymbol{w})$ is present. In this section, to remove this assumption, we design a variant of the proximal ADMM by smoothing the regularizer $g(\boldsymbol{w})$. We employ the Moreau envelope to smooth $g(\boldsymbol{w})$ and replace $g(\boldsymbol{w})$ in Algorithm 1 with $M_{g,\gamma}(\boldsymbol{w})$. We point out that $M_{g,\gamma}(\boldsymbol{w})$ is a $\gamma^{-1}$ weakly convex function if $0 < c\gamma \leq \frac{1}{3}$. Thus the $w$-subproblem is still strongly convex and can be solved exactly. See Appendix A.1 for proof. We assume that the sequence $\{\boldsymbol{\lambda}^k\}$ is bounded, which is much weaker than Assumption 2. We will show that within $O(1/\epsilon^4)$ iterations, $(\boldsymbol{z}^k, \tilde{\boldsymbol{w}}^k, \boldsymbol{\lambda}^k)$ is an $\epsilon$-KKT point, where $\tilde{\boldsymbol{w}}^k = \text{prox}_{g,\gamma}(\boldsymbol{w}^k)$.

To compensate for the absence of Assumption 2, we introduce the following more practical assumptions.

**Assumption 4** *We assume* $\boldsymbol{z}^k \in Im(D)\ \forall k$, *where* $Im(D) = \{D\boldsymbol{x} : \boldsymbol{x} \in \mathbb{R}^d\}$.

A stronger version of Assumption 4 is that $DD^\top = (\text{diag}(\boldsymbol{y})X)(\text{diag}(\boldsymbol{y})X)^\top = \text{diag}(\boldsymbol{y})XX^\top\text{diag}(\boldsymbol{y}) \succ 0$ [32]. It is worth noting that the full row rank property of the data matrix is often assumed in the high dimensional setting classification ($m < d$ and each entry of $\boldsymbol{y}$ belongs to $\{-1, 1\}$). Here we do not impose any assumptions on the rank of data matrix $X$.

**Assumption 5** $\nabla M_{g,\gamma}(\boldsymbol{w})$ *is bounded by* $M > 0$, *i.e.,* $\|\nabla M_{g,\gamma}(\boldsymbol{w})\| \leq M,\ \forall \boldsymbol{w} \in \mathbb{R}^n$.

Regarding nonsmooth regularizers, Assumption 5 is satisfied by weakly convex functions that are Lipschitz continuous [7], due to the fact that Lipschitz continuity implies bounded subgradients and (8). Common regularizers such as $l_1$-norm, MCP and SCAD are all Lipschitz continuous.

The following proposition establishes the relationship between $\{(\boldsymbol{z}^k, \boldsymbol{w}^k, \boldsymbol{\lambda}^k)\}$ and $\{(\boldsymbol{z}^k, \tilde{\boldsymbol{w}}^k, \boldsymbol{\lambda}^k)\}$.

**Proposition 2** *Suppose g is c-weakly convex and Assumption 3 holds. Suppose the sequence $\{(\boldsymbol{z}^k, \boldsymbol{w}^k, \boldsymbol{\lambda}^k)\}$ is produced during the iterations of Algorithm 1 with replacing $g(\boldsymbol{w})$ by $M_{g,\gamma}(\boldsymbol{w})$ and $\tilde{\boldsymbol{w}}^k = \mathrm{prox}_{g,\gamma}(\boldsymbol{w}^k)$. Then we have*

$$\mathbf{0} \in \partial\Omega(\boldsymbol{z}^{k+1}) + \boldsymbol{\lambda}^{k+1} + \rho D(\boldsymbol{w}^{k+1} - \boldsymbol{w}^k), \qquad D^T\boldsymbol{\lambda}^{k+1} + r(\boldsymbol{w}^k - \boldsymbol{w}^{k+1}) \in \partial g(\tilde{\boldsymbol{w}}^{k+1}). \quad (13)$$

Before presenting our main results, we introduce a Lyapunov function that plays a significant role in our analysis:

$$\Phi^k = L_\rho(\boldsymbol{w}^k, \boldsymbol{z}^k, \boldsymbol{\lambda}^k) + \frac{2r^2}{\sigma\rho}\|\boldsymbol{w}^k - \boldsymbol{w}^{k-1}\|^2.$$

The following lemma demonstrates the sufficient decrease property for $\Phi^k$.

**Lemma 1** *Let $r > \gamma^{-1}$. Under Assumptions 1, 3, 4 and 5, if $0 < c\gamma \leq \frac{1}{3}$, then we have*

$$\Phi^k - \Phi^{k+1} \geq \left(\frac{2r - \gamma^{-1}}{2} - \frac{4r^2}{\sigma\rho} - \frac{2}{\sigma\rho\gamma^2}\right)\|\boldsymbol{w}^{k+1} - \boldsymbol{w}^k\|^2 + \frac{1}{\rho}\|\boldsymbol{\lambda}^{k+1} - \boldsymbol{\lambda}^k\|^2. \quad (14)$$

The sufficient decrease property for the Lyapunov function is crucial in the proof of nonconvex ADMM. Using Lemma 1, we can control $\|\boldsymbol{w}^{k+1} - \boldsymbol{w}^k\|$ and $\|\boldsymbol{\lambda}^{k+1} - \boldsymbol{\lambda}^k\|$ through the decrease of the Lyapunov function. We are now ready to present the main result of this section.

**Theorem 2** *Suppose $\{\boldsymbol{\lambda}^k\}$ is bounded. Set $\gamma = \epsilon \leq \frac{1}{3c}$, $\rho = \frac{C_1}{\epsilon}$, $r = \frac{C_2}{\epsilon}$, where $C_1$ and $C_2$ are constants such that $C_2 > 1$ and $C_1 > \frac{8C_2^2 + \frac{1}{3c} + 4}{\sigma(2C_2 - 1)}$. Under the same assumptions in Lemma 1, Algorithm 1 with replacing $g(\boldsymbol{w})$ by $M_{g,\gamma}(\boldsymbol{w})$ finds an $\epsilon$-KKT point $(\boldsymbol{z}^{k+1}, \tilde{\boldsymbol{w}}^{k+1}, \boldsymbol{\lambda}^{k+1})$ within $O(1/\epsilon^4)$ iterations, i.e.,*

$$\mathrm{dist}\left(-\boldsymbol{\lambda}^{k+1}, \partial\Omega\left(\boldsymbol{z}^{k+1}\right)\right) \leq \epsilon, \quad \mathrm{dist}\left(D^T\boldsymbol{\lambda}^{k+1}, \partial g\left(\tilde{\boldsymbol{w}}^{k+1}\right)\right) \leq \epsilon, \quad \|\boldsymbol{z}^{k+1} - D\tilde{\boldsymbol{w}}^{k+1}\| \leq \epsilon.$$

## 5 Numerical Experiment

In this section, we perform binary classification experiments to illustrate both the robustness and the extensive applicability of our proposed algorithm.

When using the logistic loss or hinge loss as individual loss, the objective function of problem (1) can be rewritten as $\sum_{i=1}^n \sigma_i \log\left(1 + \exp\left(z_{[i]}\right)\right) + g(\boldsymbol{w})$ or $\sum_{i=1}^n \sigma_i \left[1 + z_{[i]}\right]_+ + g(\boldsymbol{w})$, where $\boldsymbol{z} = -\boldsymbol{y} \odot (X\boldsymbol{w})$, and $g(\boldsymbol{w}) = \frac{\mu}{2}\|\boldsymbol{w}\|_2^2$ or $g(\boldsymbol{w}) = \frac{\mu}{2}\|\boldsymbol{w}\|_1$.

We point out some key settings:

1) For the spectral risk measures, we use the logistic loss and hinge loss as individual loss, and use $\ell_1$ and $\ell_2$ regularization with $\mu$ taken as $10^{-2}$.
2) For the average of ranked range aggregate loss, we use the logistic loss and hinge loss as individual loss and use $\ell_2$ regularization with $\mu = 10^{-4}$.
3) For the empirical human risk minimization, we use the logistic loss as individual loss, and use $\ell_2$ regularization with $\mu = 10^{-2}$.

As shown in Section 2.1, we can apply our proposed algorithm to a variety of frameworks such as the spectral risk measure (SRM), the average of ranked-range (AoRR) aggregate loss, and the empirical human risk minimization (EHRM). We compare our algorithm with LSVRG, SGD, and DCA: LSVRG(NU) denotes LSVRG without uniformity, and LSVRG(U) denotes LSVRG with uniformity, as detailed in [36]; SGD denotes the stochastic subgradient method in [36]; DCA denotes the difference-of-convex algorithm in [40]; sADMM refers to the smoothed version of ADMM.

The details of our algorithm setting, more experiment settings and detailed information for each dataset are provided in Appendix B. Additional experiments with synthetic datasets are presented in Appendix C.

## 5.1 Spectral Risk Measures

In this section, we select $\sigma(t) \equiv 1$ in (3) to generate the empirical risk minimization (ERM) and $\sigma(t) = 1_{[q,1]}(t)/(1-q)$ to generate superquantile, the latter having garnered significant attention in the domains of finance quantification and machine learning [30].

**Setting.** For the learning rates of SGD and LSVRG, we use grid search in the range $\{10^{-2}, 10^{-3}, 10^{-4}, 1/N_{\text{samples}}, 1/(N_{\text{samples}} \times N_{\text{features}})\}$, which is standard for stochastic algorithms, and then we repeat the experiments five times with different random seeds.

**Results.** Tables 1 and 2 present the mean and standard deviation of the results in different regularizations. The first row for each dataset represents the objective value of problem (1) and the second row shows test accuracy. Bold numbers indicate significantly better results. Tables 1 and 2 show that our ADMM algorithm outperforms existing methods in terms of objective values for most instances and has comparable test accuracy. In addition, we highlight the necessity of reselecting the learning rate for SGD and LSVRG algorithms, as the individual loss and dataset undergo modifications—a highly intricate procedure. Conversely, our proposed algorithm circumvents the need for any adjustments in this regard.

Table 1: Results in real data with SRM superquantile framework and $\ell_2$ regularization.

| Datasets | Logistic Loss | | | | Hinge Loss | | | |
|---|---|---|---|---|---|---|---|---|
| | ADMM | LSVRG(NU) | LSVRG(U) | SGD | ADMM | LSVRG(NU) | LSVRG(U) | SGD |
| SVMguide | 0.6522 (0.0058) | 0.6522 (0.0058) | 0.6522 (0.0058) | 0.6523 (0.0058) | 0.7724 (0.0214) | 0.7724 (0.0214) | 0.7724 (0.0214) | 0.7738 (0.0219) |
| | 0.9566 (0.0035) | 0.9563 (0.0038) | 0.9561(0.0036) | 0.9566 (0.0032) | 0.9568 (0.0033) | 0.9568 (0.0033) | 0.9568 (0.0033) | 0.9574 (0.0035) |
| AD | 0.1539 (0.0092) | 0.1538 (0.0092) | 0.1538 (0.0091) | 0.1560 (0.0114) | **0.0787 (0.0132)** | 0.0852 (0.0129) | 0.0833 (0.0134) | 0.0815 (0.0137) |
| | 0.9629 (0.0071) | 0.9633 (0.0070) | 0.9633 (0.0070) | 0.9631 (0.0067) | 0.9521 (0.0112) | 0.9559 (0.0099) | 0.9578 (0.0098) | 0.9576 (0.0097) |

Table 2: Results in real data with SRM superquantile framework and $\ell_1$ regularization.

| Datasets | Logistic Loss | | | | | Hinge Loss | | | | |
|---|---|---|---|---|---|---|---|---|---|---|
| | ADMM | sADMM | LSVRG(NU) | LSVRG(U) | SGD | ADMM | sADMM | LSVRG(NU) | LSVRG(U) | SGD |
| SVMguide | 0.6280 (0.0129) | 0.6280 (0.0129) | 0.6280 (0.0129) | 0.6280 (0.0129) | 0.6282 (0.0128) | 0.6789 (0.0273) | 0.6789 (0.0273 ) | 0.9121 (0.0133) | 0.6789 (0.0273) | 0.8569 (0.0210) |
| | 0.9561 (0.0037) | 0.9561 (0.0037) | 0.9563 (0.0039) | 0.9561 (0.0037) | 0.9561 (0.0038) | 0.9566 (0.0036) | 0.9566 (0.0036) | 0.9558 (0.0039) | 0.9563 (0.0035) | 0.9555 (0.0040) |
| AD | **0.2601 (0.0079)** | 0.2602 (0.0079) | 0.2614 (0.0080) | 0.2614 (0.0080) | 0.2641 (0.0105) | **0.1352 (0.0094)** | 0.1370 (0.0096) | 0.1457 (0.0110) | 0.1449 (0.0100) | 0.1402 (0.0101) |
| | 0.9673 (0.0054) | 0.9673 (0.0054) | 0.9680 (0.0054) | 0.9680 (0.0054) | 0.9673 (0.0039) | 0.9589 (0.0060) | 0.9589 (0.0060) | 0.9644 (0.0045) | 0.9639 (0.0039) | 0.9635 (0.0023) |

Figure 1 illustrates the relationship between time and sub-optimality of each algorithm in the ERM framework with logistic loss, which is a convex problem whose global optimum can be achieved. The learning rate of the other algorithms is the best one selected by the same method. The Figure 1 depicts that ADMM exhibits significantly faster convergence towards the minimum in comparison to the other algorithms. Conversely, although sADMM fails to achieve the accuracy level of ADMM, it nevertheless demonstrates accelerated convergence when compared to other algorithms.

## 5.2 Empirical Human Risk Minimization

Since [31] utilizes gradient-based algorithms akin to those in [36], we compare our algorithm with the one referenced in Section 5.1.

**Results.** The mean and standard deviation of the results are tabulated in Table 3. SPD, DI, EOD, AOD, TI, and FNRD are fairness metrics detailed in Appendix B.6. The bold numbers highlight the superior result. It is evident that our proposed algorithm outperforms the other algorithms in terms of objective values and test accuracy, and also shows enhanced performance across almost all fairness metrics.

## 5.3 Average of Ranked Range Aggregate Loss

As mentioned in Section 2.1, DCA can be used to solve for AoRR aggregate loss minimization [27]. Therefore, we compare our algorithm with the DCA algorithm in this section.

**Results.** Table 4 displays the mean and standard deviation of both the objective value and time results. Given that this is a nonconvex problem, the objective value may not be the optimal value.

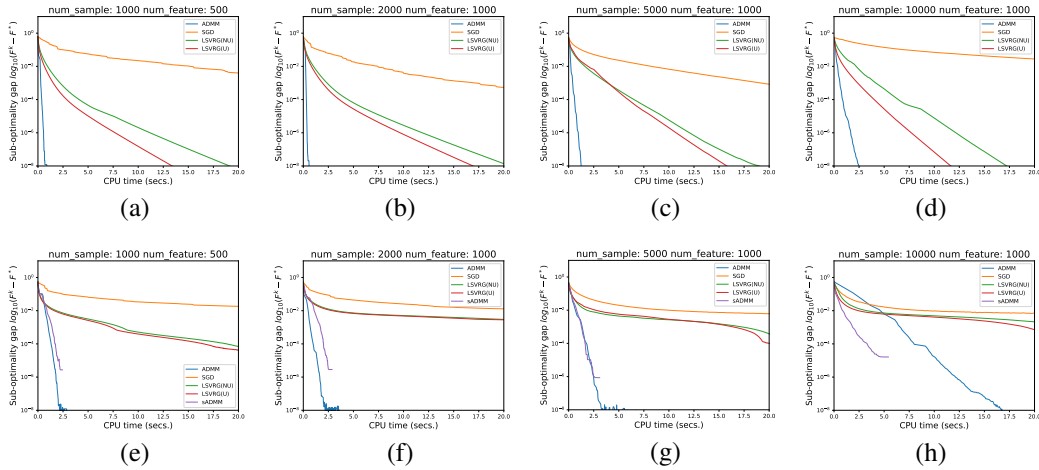

Figure 1: Time vs. Sub-optimality gap in synthetic datasets with ERM framework. (a-d) for $\ell_2$ regularization, and (e-f) for $\ell_1$ regularization. Sub-optimality is defined as $F^k - F^*$, where $F^k$ represents the objective function value at the $k$-th iteration or epoch and $F^*$ denotes the minimum value obtained by all algorithms. Plots are truncated when $F^k - F^* < 10^{-8}$.

Table 3: Comparison with EHRM framework. 'objval' denotes the objective value of problem (1).

|          | ADMM               | LSVRG(NU)       | LSVRG(U)        | SGD             |      | ADMM              | LSVRG(NU)        | LSVRG(U)         | SGD               |
|----------|--------------------|-----------------|-----------------|-----------------|------|-------------------|------------------|------------------|-------------------|
| objval   | **0.4955 (0.0045)** | 0.5383 (0.0022) | 0.5384 (0.0023) | 0.6093 (0.0013) | EOD  | -0.0521 (0.0160)  | -0.0238 (0.0114) | -0.0231 (0.0110) | **-0.0160(0.0103)** |
| Accuracy | **0.7759 (0.0076)** | 0.7445 (0.0050) | 0.7448 (0.0054) | 0.6334 (0.0205) | AOD  | **0.0130 (0.0090)** | 0.0449 (0.0199)  | 0.0450 (0.0196)  | 0.0426 (0.0078)   |
| SPD      | **0.0300 (0.0071)** | 0.0585 (0.0161) | 0.0587 (0.0158) | 0.0459 (0.0050) | TI   | **0.0838 (0.0018)** | 0.0933 (0.0018)  | 0.0934 (0.0018)  | 0.1062 (0.0052)   |
| DI       | **1.0533 (0.0126)** | 1.1034 (0.0300) | 1.104 (0.0297)  | 1.0586 (0.0065) | FNRD | 0.0521 (0.0160)   | 0.0238 (0.0114)  | 0.0231 (0.0110)  | **0.0160 (0.0103)** |

Nevertheless, as evidenced in Table 4, our proposed algorithm achieves better objective value in a shorter time for all the instances.

Table 4: Comparison with AoRR framework. 'objval' denotes the objective value of problem (1).

| Datasets |          | Monks              |                 | Australian         |                 | Phoneme            |                 | Titanic            |                 | Splice             |                 |
|----------|----------|--------------------|-----------------|--------------------|-----------------|--------------------|-----------------|--------------------|-----------------|--------------------|-----------------|
|          |          | ADMM               | DCA             | ADMM               | DCA             | ADMM               | DCA             | ADMM               | DCA             | ADMM               | DCA             |
| *Logistic* | objval | **0.0025 (0.0001)** | 0.5857 (0.1258) | **0.0011 (0.0001)** | 0.6919 (0.0003) | **0.0031 (0.0001)** | 0.685 (0.0090)  | **0.0027 (0.0001)** | 0.6912 (0.0003) | **0.0018 (0.0001)** | 0.6914 (0.0003) |
|          | time (s) | **3.15 (0.11)**    | 46.73 (0.29)    | **3.54 (0.34)**    | 53.85 (0.98)    | **55.36 (5.89)**   | 116.24 (27.16)  | **40.98 (9.36)**   | 51.8 (0.21)     | **17.58 (0.37)**   | 63.91 (5.41)    |
| *Hinge*  | objval   | **0.0093 (0.0017)** | 0.8556 (0.1701) | **0.0017 (0.0010)** | 0.9949 (0.0012) | **0.0060 (0.0005)** | 0.8842 (0.1657) | **0.0127 (0.0021)** | 0.9923 (0.0013) | **0.0038 (0.0005)** | 0.9931 (0.0013) |
|          | time (s) | **1.05 (0.02)**    | 23.80 (0.13)    | **1.62 (0.05)**    | 24.15 (0.07)    | **20.28 (3.31)**   | 39.09 (1.60)    | **7.22 (1.26)**    | 14.53 (1.53)    | **9.12 (1.42)**    | 114.52 (26.94)  |

# 6  Conclusion

This paper considers rank-based loss optimization with monotonically increasing loss functions and weakly convex regularizers. We propose a unified ADMM framework for rank-based loss minimization. Notably, one subproblem of the ADMM is solved efficiently by the PAVA. Numerical experiments illustrate the outperformance of our proposed algorithm, with all three practical frameworks delivering satisfactory results. We also point out some limitations of our algorithm. To effectively utilize our algorithm, individual losses must exhibit monotonicity, as this allows us to use the PAVA to solve subproblems. If the sample size is large, the PAVA's computational efficiency may be hindered, potentially limiting its overall effectiveness. Future work may explore a variant using mini-batch samples, potentially improving early-stage optimization performance and overall computational efficiency.

**Acknowledgement**    Rujun Jiang is partly supported by NSFC 12171100 and Natural Science Foundation of Shanghai 22ZR1405100.

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

# Appendix

## A Proofs

### A.1 Proofs for properties of Moreau envelope

**Lemma 2** *Let $g : \mathbb{R}^n \to \mathbb{R} \cup \{\infty\}$ be a c-weakly convex function. For $0 < c\gamma \leq \frac{1}{3}$, the Moreau envelope $M_{g,\gamma}$ of $g$ is $\gamma^{-1}$-weakly convex.*

#### A.1.1 Proof of Lemma 2

**Lemma 3** *Let $g : \mathbb{R}^n \to \mathbb{R} \cup \{\infty\}$. Then g is c-weakly convex if and only if the following holds*

$$\langle \boldsymbol{v} - \boldsymbol{w}, \boldsymbol{x} - \boldsymbol{y} \rangle \geq -c\|\boldsymbol{x} - \boldsymbol{y}\|^2,$$

*for $\boldsymbol{x}, \boldsymbol{y} \in \mathbb{R}^n$, $\boldsymbol{v} \in \partial g(\boldsymbol{x})$, $\boldsymbol{w} \in \partial g(\boldsymbol{y})$.*

*Proof:* See [16, Lemma 2.1]. ∎

**Lemma 4** *Suppose $g(\boldsymbol{w})$ is c-weakly convex. If $0 < c\gamma \leq \frac{1}{3}$, then we have*

$$\| \mathrm{prox}_{g,\gamma}(\boldsymbol{x}) - \mathrm{prox}_{g,\gamma}(\boldsymbol{y}) \|^2 \leq 3\|\boldsymbol{y} - \boldsymbol{x}\|^2.$$

*Proof:* Suppose $\boldsymbol{x}, \boldsymbol{y} \in \mathbb{R}^n$. By (8) and Lemma 3 we have

$$\begin{aligned}
&- c\| \mathrm{prox}_{g,\gamma}(\boldsymbol{x}) - \mathrm{prox}_{g,\gamma}(\boldsymbol{y}) \|^2 \\
&\leq \gamma^{-1} \langle (\boldsymbol{y} - \mathrm{prox}_{g,\gamma}(\boldsymbol{y})) - (\boldsymbol{x} - \mathrm{prox}_{g,\gamma}(\boldsymbol{x})), \mathrm{prox}_{g,\gamma}(\boldsymbol{y}) - \mathrm{prox}_{g,\gamma}(\boldsymbol{x}) \rangle \\
&= \frac{1}{2\gamma} [ -\|\boldsymbol{y} - \mathrm{prox}_{g,\gamma}(\boldsymbol{y}) - \boldsymbol{x} + \mathrm{prox}_{g,\gamma}(\boldsymbol{x})\|^2 - \| \mathrm{prox}_{g,\gamma}(\boldsymbol{y}) - \mathrm{prox}_{g,\gamma}(\boldsymbol{x}) \|^2 \\
&\quad + \|\boldsymbol{x} - \boldsymbol{y}\|^2 ].
\end{aligned}$$

Thus we have

$$(1 - 2c\gamma)\| \mathrm{prox}_{g,\gamma}(\boldsymbol{y}) - \mathrm{prox}_{g,\gamma}(\boldsymbol{x}) \|^2 \leq -\|\boldsymbol{y} - \mathrm{prox}_{g,\gamma}(\boldsymbol{y}) - \boldsymbol{x} + \mathrm{prox}_{g,\gamma}(\boldsymbol{x})\|^2 + \|\boldsymbol{x} - \boldsymbol{y}\|^2$$
$$\leq \|\boldsymbol{x} - \boldsymbol{y}\|^2.$$

Note that $(1 - 2c\gamma)^{-1} \leq 3$ when $0 < c\gamma \leq \frac{1}{3}$. Thus we have

$$\| \mathrm{prox}_{g,\gamma}(\boldsymbol{y}) - \mathrm{prox}_{g,\gamma}(\boldsymbol{x}) \|^2 \leq (1 - 2c\gamma)^{-1}\|\boldsymbol{x} - \boldsymbol{y}\|^2 \leq 3\|\boldsymbol{x} - \boldsymbol{y}\|^2.$$

∎

**Proof of Lemma 2**

*Proof:* For any $\boldsymbol{x}, \boldsymbol{y} \in \mathbb{R}^n$, we have

$$\begin{aligned}
&\langle \nabla M_{g,\gamma}(\boldsymbol{x}) - \nabla M_{g,\gamma}(\boldsymbol{y}), \boldsymbol{x} - \boldsymbol{y} \rangle \\
&= \gamma^{-1} \langle \boldsymbol{x} - \mathrm{prox}_{g,\gamma}(\boldsymbol{x}) - \boldsymbol{y} + \mathrm{prox}_{g,\gamma}(\boldsymbol{y}), \boldsymbol{x} - \boldsymbol{y} \rangle \\
&= \gamma^{-1} \|\boldsymbol{x} - \boldsymbol{y}\|^2 + \gamma^{-1} \langle \mathrm{prox}_{g,\gamma}(\boldsymbol{y}) - \mathrm{prox}_{g,\gamma}(\boldsymbol{x}), \boldsymbol{x} - \boldsymbol{y} \rangle \\
&\geq \gamma^{-1} \|\boldsymbol{x} - \boldsymbol{y}\|^2 - \frac{1}{2\gamma}\|\boldsymbol{x} - \boldsymbol{y}\|^2 - \frac{1}{2\gamma}\| \mathrm{prox}_{g,\gamma}(\boldsymbol{x}) - \mathrm{prox}_{g,\gamma}(\boldsymbol{y}) \|^2 \\
&\geq \frac{1}{2\gamma}\|\boldsymbol{x} - \boldsymbol{y}\|^2 - \frac{3}{2\gamma}\|\boldsymbol{x} - \boldsymbol{y}\|^2 \\
&= -\frac{1}{\gamma}\|\boldsymbol{x} - \boldsymbol{y}\|^2,
\end{aligned}$$

where the first equality follows from (8) and the last inequality follows from Lemma 4. Thus $g$ is $\gamma^{-1}$-weakly convex by Lemma 3. ∎

## A.2 Proofs for ADMM in Section 3

### A.2.1 Proof of Proposition 1

*Proof:* First we define $h_{[a,b]}(z) = \sum_{i=a}^{b} \theta_i(z)$. Due to convexity, we have

$$0 \in \partial h_{[m,n]}(v_{[m,n]}) \quad \text{and} \quad 0 \in \partial h_{[n+1,p]}(v_{[n+1,p]}).$$

Noting that $v_{[m,n]} > v_{[n+1,p]}$, due to the strong convexity of $h$, we further have

$$\partial h_{[m,n]}(v_{[n+1,p]}) < 0, \quad \text{and} \quad \partial h_{[n+1,p]}(v_{[m,n]}) > 0,$$

where we use the convention that a set $A > 0$ (or $< 0$) denotes that all elements $g \in A$ satisfies $g > 0$ (or $< 0$). Therefore we obtain that there exist $s_{[m,n]}, s_{[n+1,p]} \in \mathbb{R}$ such that

$$\partial h_{[m,p]}(v_{[m,n]}) = \partial h_{[m,n]}(v_{[m,n]}) + \partial h_{[n+1,p]}(v_{[m,n]}) \ni s_{[m,n]} > 0,$$
$$\partial h_{[m,p]}(v_{[n+1,p]}) = \partial h_{[m,n]}(v_{[n+1,p]}) + \partial h_{[n+1,p]}(v_{[n+1,p]}) \ni s_{[n+1,p]} < 0.$$

As $h_{[m,p]}$ is strongly convex in $[v_{[n+1,p]}, v_{[m,n]}]$, the subgradient $\partial h_{[m,p]}$ is strictly increasing in $[v_{[n+1,p]}, v_{[m,n]}]$. That is, $\forall x, y \in [v_{[n+1,p]}, v_{[m,n]}], \forall s_x \in \partial h_{[m,p]}(x), s_y \in \partial h_{[m,p]}(y), (s_x - s_y)(x - y) \geq 0$. The above facts, together with the strong convexity of $h_{[m,p]}$, the unique minimizer $v_{[m,p]}$ of $h_{[m,p]}$ must lie in the interval $[v_{[n+1,p]}, v_{[m,n]}]$. ∎

### A.2.2 Proof of Theorem 1

*Proof:* We use the extended formula for Clark generalized gradient of a sum of two functions in our proofs: $\partial(f_1 + f_2)(x) \subset \partial f_1(x) + \partial f_2(x)$ if $f_1$ and $f_2$ are finite at $x$ and $f_2$ is differentiable at $x$. The equality holds if $f_1$ is regular at $x$. [10, Theorem 2.9.8].

By Assumption 3, we have

$$
\begin{aligned}
\mathbf{0} \in \partial_z & \left( \Omega(z^{k+1}) + \frac{\rho}{2}\|z^{k+1} - Dw^k + \frac{\lambda^k}{\rho}\|^2 \right) \\
& \subset \partial\Omega(z^{k+1}) + \lambda^k + \rho(z^{k+1} - Dw^k) \\
& = \partial\Omega(z^{k+1}) + \lambda^k + \rho(z^{k+1} - Dw^{k+1}) + \rho D(w^{k+1} - w^k) \\
& = \partial\Omega(z^{k+1}) + \lambda^{k+1} + \rho D(w^{k+1} - w^k),
\end{aligned}
\tag{15}
$$

and

$$L_\rho(z^k, w^k, \lambda^k) - L_\rho(z^{k+1}, w^k, \lambda^k) \geq 0. \tag{16}$$

Here the last equality of (15) follows from $\lambda^{k+1} = \lambda^k + \rho(z^{k+1} - Dw^{k+1})$.

By Assumption 3 and $(r - c)$-strong convexity of the $w$-subproblem, we have

$$
\begin{aligned}
\mathbf{0} \in \partial_w & \left( \frac{\rho}{2}\|z^{k+1} - Dw^{k+1} + \frac{\lambda^k}{\rho}\|^2 + g(w^{k+1}) + \frac{r}{2}\|w^{k+1} - w^k\|^2 \right) \\
& = \partial g(w^{k+1}) - D^\top\lambda^k - \rho D^\top(z^{k+1} - Dw^{k+1}) + r(w^{k+1} - w^k) \\
& = \partial g(w^{k+1}) - D^\top\lambda^{k+1} + r(w^{k+1} - w^k),
\end{aligned}
\tag{17}
$$

and

$$L_\rho(z^{k+1}, w^k, \lambda^k) - L_\rho(z^{k+1}, w^{k+1}, \lambda^k) \geq \frac{2r - c}{2}\|w^{k+1} - w^k\|^2. \tag{18}$$

The second equality of (17) is due to the fact that weakly convex functions are regular [47, Proposition 4.5]. By the dual update we have

$$
\begin{aligned}
& L_\rho(z^{k+1}, w^{k+1}, \lambda^k) - L_\rho(z^{k+1}, w^{k+1}, \lambda^{k+1}) \\
& = \left( \lambda^k - \lambda^{k+1} \right)^\top \left( z^{k+1} - Dw^{k+1} \right) \\
& = -\frac{1}{\rho}\|\lambda^{k+1} - \lambda^k\|^2.
\end{aligned}
\tag{19}
$$

Summing (16), (18) and (19) we obtain that

$$L_\rho(\boldsymbol{z}^k, \boldsymbol{w}^k, \boldsymbol{\lambda}^k) - L_\rho(\boldsymbol{z}^{k+1}, \boldsymbol{w}^{k+1}, \boldsymbol{\lambda}^{k+1}) \geq \frac{2r-c}{2}\|\boldsymbol{w}^{k+1} - \boldsymbol{w}^k\|^2 - \frac{1}{\rho}\|\boldsymbol{\lambda}^k - \boldsymbol{\lambda}^{k+1}\|^2. \quad (20)$$

By Assumptions 1 and 2 we have that

$$L_\rho(\boldsymbol{z}^k, \boldsymbol{w}^k, \boldsymbol{\lambda}^k) = \Omega(\boldsymbol{z}^k) + g(\boldsymbol{w}^k) + (\boldsymbol{\lambda}^k)^\top(\boldsymbol{z}^k - D\boldsymbol{w}^k) + \frac{\rho}{2}\|\boldsymbol{z}^k - D\boldsymbol{w}^k\|^2$$

$$= \Omega(\boldsymbol{z}^k) + g(\boldsymbol{w}^k) + \frac{\rho}{2}\left\|\boldsymbol{z}^k - D\boldsymbol{w}^k + \frac{\boldsymbol{\lambda}^k}{\rho}\right\|^2 - \frac{\|\boldsymbol{\lambda}^k\|^2}{2\rho} \quad (21)$$

$$\geq -\frac{\|\boldsymbol{\lambda}^k\|^2}{2\rho} > -\infty.$$

So $L_\rho(\boldsymbol{z}^k, \boldsymbol{w}^k, \boldsymbol{\lambda}^k)$ is bounded below by some $L^\star$. Moreover, $\sum_{k=1}^K \|\boldsymbol{\lambda}^{k+1} - \boldsymbol{\lambda}^k\|^2 \leq \sum_{k=1}^\infty \|\boldsymbol{\lambda}^{k+1} - \boldsymbol{\lambda}^k\|^2 < \infty$ for $\forall K \geq 1$.

Let $\hat{L} := L_\rho(\boldsymbol{z}^1, \boldsymbol{w}^1, \boldsymbol{\lambda}^1) - L^\star + \sum_{k=1}^\infty \frac{2}{\rho}\|\boldsymbol{\lambda}^{k+1} - \boldsymbol{\lambda}^k\|^2 < \infty$. Then telescoping (20) from $k = 1$ to $K$, we obtain that

$$\hat{L} \geq L_\rho(\boldsymbol{z}^1, \boldsymbol{w}^1, \boldsymbol{\lambda}^1) - L_\rho(\boldsymbol{z}^K, \boldsymbol{w}^K, \boldsymbol{\lambda}^K) + \sum_{k=1}^K \frac{2}{\rho}\|\boldsymbol{\lambda}^{k+1} - \boldsymbol{\lambda}^k\|^2$$

$$\geq \sum_{k=1}^K \frac{2r-c}{2}\|\boldsymbol{w}^{k+1} - \boldsymbol{w}^k\|^2 + \sum_{k=1}^K \frac{1}{\rho}\|\boldsymbol{\lambda}^{k+1} - \boldsymbol{\lambda}^k\|^2. \quad (22)$$

This implies that

$$\min_{k \leq K} \frac{2r-c}{2}\|\boldsymbol{w}^{k+1} - \boldsymbol{w}^k\|^2 + \frac{1}{\rho}\|\boldsymbol{\lambda}^{k+1} - \boldsymbol{\lambda}^k\|^2 \leq \frac{\hat{L}}{K}.$$

Letting $k = \arg\min_{i \leq K} \frac{2r-c}{2}\|\boldsymbol{w}^{i+1} - \boldsymbol{w}^i\|^2 + \frac{1}{\rho}\|\boldsymbol{\lambda}^{i+1} - \boldsymbol{\lambda}^i\|^2$, we have

$$\|\boldsymbol{w}^{k+1} - \boldsymbol{w}^k\| \leq \sqrt{\frac{2\hat{L}}{K(2r-c)}},$$

$$\|\boldsymbol{\lambda}^{k+1} - \boldsymbol{\lambda}^k\| \leq \sqrt{\frac{\rho\hat{L}}{K}}.$$

Letting $K = \frac{1}{\epsilon^2}$, by (15) and (17) we further have

$$\text{dist}\left(-\boldsymbol{\lambda}^{k+1}, \partial\Omega\left(\boldsymbol{z}^{k+1}\right)\right) \leq \rho\|D\|\|\boldsymbol{w}^{k+1} - \boldsymbol{w}^k\| \leq \rho\|D\|\sqrt{\frac{2\hat{L}}{K(2r-c)}} = O(\epsilon),$$

$$\text{dist}\left(D^T\boldsymbol{\lambda}^{k+1}, \partial g\left(\boldsymbol{w}^{k+1}\right)\right) \leq r\|\boldsymbol{w}^{k+1} - \boldsymbol{w}^k\| \leq r\sqrt{\frac{2\hat{L}}{K(2r-c)}} = O(\epsilon),$$

$$\|\boldsymbol{z}^{k+1} - D\boldsymbol{w}^{k+1}\| = \frac{1}{\rho}\|\boldsymbol{\lambda}^{k+1} - \boldsymbol{\lambda}^k\| \leq \sqrt{\frac{\hat{L}}{\rho K}} = O(\epsilon).$$

∎

The proof of Theorem 1 is adapted from [33, Theorem 4.1].

### A.3 Proofs for smoothed ADMM in Section 4

#### A.3.1 Proof of Proposition 2

*Proof:* By Assumption 3, similar to (15), we have

$$\mathbf{0} \in \partial\Omega(\boldsymbol{z}^{k+1}) + \boldsymbol{\lambda}^k + \rho(\boldsymbol{z}^{k+1} - D\boldsymbol{w}^k)$$
$$= \partial\Omega(\boldsymbol{z}^{k+1}) + \boldsymbol{\lambda}^k + \rho(\boldsymbol{z}^{k+1} - D\boldsymbol{w}^{k+1}) + \rho D(\boldsymbol{w}^{k+1} - \boldsymbol{w}^k)$$
$$= \partial\Omega(\boldsymbol{z}^{k+1}) + \boldsymbol{\lambda}^{k+1} + \rho D(\boldsymbol{w}^{k+1} - \boldsymbol{w}^k),$$

where the last equality follows from $\boldsymbol{\lambda}^{k+1} = \boldsymbol{\lambda}^k + \rho(\boldsymbol{z}^{k+1} - D\boldsymbol{w}^{k+1})$.

By the first order condition of the $\boldsymbol{w}$-subproblem and Assumption 3, we have

$$\mathbf{0} = \nabla M_{g,\gamma}(\boldsymbol{w}^{k+1}) - D^\top\boldsymbol{\lambda}^k - \rho D^\top(\boldsymbol{z}^{k+1} - D\boldsymbol{w}^{k+1}) + r(\boldsymbol{w}^{k+1} - \boldsymbol{w}^k)$$
$$= \nabla M_{g,\gamma}(\boldsymbol{w}^{k+1}) - D^\top\boldsymbol{\lambda}^{k+1} + r(\boldsymbol{w}^{k+1} - \boldsymbol{w}^k). \tag{23}$$

Note that $\nabla M_{g,\gamma}(\boldsymbol{w}^{k+1}) \in \partial g\left(\text{prox}_{g,\gamma}(\boldsymbol{w}^{k+1})\right) = \partial g(\tilde{\boldsymbol{w}}^{k+1})$. This completes the proof. ■

#### A.3.2 Proof of Lemma 1

**Lemma 5** *Under Assumption 4, if $0 < c\gamma \leq \frac{1}{2}$, then for $\forall k = 0, 1, \ldots$, we have*

$$\|\boldsymbol{\lambda}^{k+1} - \boldsymbol{\lambda}^k\|^2 \leq \sigma^{-1}(\gamma^{-2} + r^2)\|\boldsymbol{w}^{k+1} - \boldsymbol{w}^k\|^2 + \sigma^{-1}r^2\|\boldsymbol{w}^{k-1} - \boldsymbol{w}^k\|^2,$$

*where $\sigma$ is the smallest positive eigenvalue of $DD^\top$.*

*Proof:* By Assumption 4 we have $\boldsymbol{\lambda}^{k+1} - \boldsymbol{\lambda}^k = \rho(\boldsymbol{z}^{k+1} - D\boldsymbol{w}^{k+1}) \in Im(D)$. Then we obtain that

$$\|\boldsymbol{\lambda}^k - \boldsymbol{\lambda}^{k+1}\|^2 \leq \sigma^{-1}\|D^\top\left(\boldsymbol{\lambda}^k - \boldsymbol{\lambda}^{k+1}\right)\|^2$$

$$\leq \sigma^{-1}\|\nabla M_{g,\gamma}(\boldsymbol{w}^{k+1}) - \nabla M_{g,\gamma}(\boldsymbol{w}^k)\|^2 + \sigma^{-1}r^2\|\boldsymbol{w}^k - \boldsymbol{w}^{k+1}\|^2 + \sigma^{-1}r^2\|\boldsymbol{w}^{k-1} - \boldsymbol{w}^k\|^2$$

$$\leq \sigma^{-1}(\gamma^{-2} + r^2)\|\boldsymbol{w}^{k+1} - \boldsymbol{w}^k\|^2 + \sigma^{-1}r^2\|\boldsymbol{w}^{k-1} - \boldsymbol{w}^k\|^2,$$

where the second inequality follows from (23) and the last inequality follows from the fact that $M_{g,\gamma}$ has $\gamma^{-1}$ Lipschitz continuous gradient when $0 < c\gamma \leq \frac{1}{2}$ [7]. ■

**Proof of Lemma 1**

*Proof:* By Assumption 3 we have

$$L_\rho(\boldsymbol{z}^k, \boldsymbol{w}^k, \boldsymbol{\lambda}^k) - L_\rho(\boldsymbol{z}^{k+1}, \boldsymbol{w}^k, \boldsymbol{\lambda}^k) \geq 0. \tag{24}$$

Since $0 < c\gamma \leq \frac{1}{3}$, $M_{g,\gamma}(\boldsymbol{w})$ is $\gamma^{-1}$-weakly convex by Lemma 2. By the optimality of $\boldsymbol{w}^{k+1}$ and $(r - \gamma^{-1})$-strong convexity of the $\boldsymbol{w}$-subproblem we have

$$L_\rho(\boldsymbol{z}^{k+1}, \boldsymbol{w}^k, \boldsymbol{\lambda}^k) - L_\rho(\boldsymbol{z}^{k+1}, \boldsymbol{w}^{k+1}, \boldsymbol{\lambda}^k) \geq \frac{2r - \gamma^{-1}}{2}\|\boldsymbol{w}^{k+1} - \boldsymbol{w}^k\|^2. \tag{25}$$

By the update of dual variable

$$L_\rho(\boldsymbol{z}^{k+1}, \boldsymbol{w}^{k+1}, \boldsymbol{\lambda}^k) - L_\rho(\boldsymbol{z}^{k+1}, \boldsymbol{w}^{k+1}, \boldsymbol{\lambda}^{k+1}) = -\frac{1}{\rho}\|\boldsymbol{\lambda}^{k+1} - \boldsymbol{\lambda}^k\|^2. \tag{26}$$

Summing (24), (25) and (26) we have

$$L_\rho(\boldsymbol{z}^k, \boldsymbol{w}^k, \boldsymbol{\lambda}^k) - L_\rho(\boldsymbol{z}^{k+1}, \boldsymbol{w}^{k+1}, \boldsymbol{\lambda}^{k+1}) \geq \frac{2r - \gamma^{-1}}{2}\|\boldsymbol{w}^{k+1} - \boldsymbol{w}^k\|^2 - \frac{1}{\rho}\|\boldsymbol{\lambda}^{k+1} - \boldsymbol{\lambda}^k\|^2. \tag{27}$$

Finally, we obtain that

$$\Phi^k - \Phi^{k+1}$$

$$= L_\rho(\boldsymbol{z}^k, \boldsymbol{w}^k, \boldsymbol{\lambda}^k) - L_\rho(\boldsymbol{z}^{k+1}, \boldsymbol{w}^{k+1}, \boldsymbol{\lambda}^{k+1}) + \frac{2r^2}{\rho\sigma}\|\boldsymbol{w}^{k-1} - \boldsymbol{w}^k\|^2 - \frac{2r^2}{\rho\sigma}\|\boldsymbol{w}^k - \boldsymbol{w}^{k+1}\|^2$$

$$\geq (\frac{2r - \gamma^{-1}}{2} - \frac{2r^2}{\rho\sigma})\|\boldsymbol{w}^{k+1} - \boldsymbol{w}^k\|^2 + \frac{2r^2}{\rho\sigma}\|\boldsymbol{w}^k - \boldsymbol{w}^{k-1}\|^2 + (-\frac{2}{\rho} + \frac{1}{\rho})\|\boldsymbol{\lambda}^{k+1} - \boldsymbol{\lambda}^k\|^2$$

$$\geq (\frac{2r - \gamma^{-1}}{2} - \frac{4r^2}{\sigma\rho} - \frac{2}{\sigma\rho\gamma^2})\|\boldsymbol{w}^{k+1} - \boldsymbol{w}^k\|^2 + \frac{1}{\rho}\|\boldsymbol{\lambda}^{k+1} - \boldsymbol{\lambda}^k\|^2,$$

where the first inequality follows from (27) and the second inequality follows from Lemma 5. ■

### A.3.3 Proof of Theorem 2

*Proof:* First note that $0 < c\gamma \le \frac{1}{3}$, so Lemma 1 holds. Let $\mathcal{E}_0 := \frac{1}{3c}$ and $\mathcal{E}_1 := \frac{r-\gamma^{-1}}{2} - \frac{4r^2}{\sigma\rho} - \frac{2}{\sigma\rho\gamma^2}$. Using the definitions of $\gamma$, $\rho$ and $r$ we have

$$
\begin{aligned}
\mathcal{E}_1 &= \frac{2r - \gamma^{-1}}{2} - \frac{4r^2}{\sigma\rho} - \frac{2}{\sigma\rho\gamma^2} \\
&= \frac{1}{2\sigma\rho\gamma^2}\left(2\sigma\rho r\gamma^2 - \sigma\rho\gamma - 8r^2\gamma^2 - 4\right) \\
&= \frac{2\sigma C_1 C_2 - \sigma C_1 - 8C_2^2 - 4}{2\sigma C_1 \epsilon} \\
&> \frac{\mathcal{E}_0}{2\sigma C_1 \epsilon} > \frac{1}{2\sigma C_1} > 0.
\end{aligned}
\tag{28}
$$

Since $\{\boldsymbol{\lambda}^k\}$ is bounded, we obtain that $L_\rho(\boldsymbol{z}^k, \boldsymbol{w}^k, \boldsymbol{\lambda}^k)$ is bounded below by a similar manner of (21) and Assumption 1. Thus $\Phi^k$ is lower bounded by some $\Phi^\star$ as well, i.e., $\Phi^k \ge \Phi^\star$, for $\forall k \ge 1$. Telescoping (14) from $k = 1$ to $K$ we have:

$$
\Phi^1 - \Phi^\star \ge \sum_{k=1}^{K} \mathcal{E}_1 \|\boldsymbol{w}^{k+1} - \boldsymbol{w}^k\|^2 + \sum_{k=1}^{K} \frac{1}{\rho}\|\boldsymbol{\lambda}^{k+1} - \boldsymbol{\lambda}^k\|^2.
$$

Thus we have:

$$
\min_{k \le K} \mathcal{E}_1 \|\boldsymbol{w}^{k+1} - \boldsymbol{w}^k\|^2 + \frac{1}{\rho}\|\boldsymbol{\lambda}^{k+1} - \boldsymbol{\lambda}^k\|^2 \le \frac{\Phi^1 - \Phi^\star}{K}.
$$

Let $k = \arg\min_{i \le K} \mathcal{E}_1 \|\boldsymbol{w}^{i+1} - \boldsymbol{w}^i\|^2 + \frac{1}{\rho}\|\boldsymbol{\lambda}^{i+1} - \boldsymbol{\lambda}^i\|^2$, we have

$$
\|\boldsymbol{w}^{k+1} - \boldsymbol{w}^k\| \le \sqrt{\frac{\Phi^1 - \Phi^\star}{K\mathcal{E}_1}},
$$

$$
\|\boldsymbol{\lambda}^{k+1} - \boldsymbol{\lambda}^k\| \le \sqrt{\frac{\rho(\Phi^1 - \Phi^\star)}{K}}.
$$

Recall $\tilde{\boldsymbol{w}}^k = \mathrm{prox}_{g,\gamma}(\boldsymbol{w}^k)$. Note that by (28), $\mathcal{E}_1^{-1}$ is upper bounded by the constant $2\sigma C_1$. Letting $K = \frac{1}{\epsilon^4}$, by Propositon 2, the definations of $r$ and $\rho$ and the above two equations, we have

$$
\mathrm{dist}\left(-\boldsymbol{\lambda}^{k+1}, \partial\Omega\left(\boldsymbol{z}^{k+1}\right)\right) \le \rho\|D\|\|\boldsymbol{w}^{k+1} - \boldsymbol{w}^k\| = O(\epsilon),
$$

$$
\mathrm{dist}\left(D^T\boldsymbol{\lambda}^{k+1}, \partial g\left(\tilde{\boldsymbol{w}}^{k+1}\right)\right) \le r\|\boldsymbol{w}^{k+1} - \boldsymbol{w}^k\| = O(\epsilon),
$$

$$
\|\boldsymbol{z}^{k+1} - D\tilde{\boldsymbol{w}}^{k+1}\| \le \|\boldsymbol{z}^{k+1} - D\boldsymbol{w}^{k+1}\| + \|D(\tilde{\boldsymbol{w}}^{k+1} - \boldsymbol{w}^{k+1})\|
$$

$$
\le \frac{1}{\rho}\|\boldsymbol{\lambda}^{k+1} - \boldsymbol{\lambda}^k\| + \gamma\|D\|M = O(\epsilon),
$$

where the second inequality follows from (8) and Assumption 5. This completes the proof. ∎

Our proof of Theorem 2 draws inspiration from [33, Theorem 4.1] and [32]. In our method, we employ the Moreau envelope $M_{g,\gamma}$ instead of $g(w)$ to obtain a smoothed version of Problem (1). However, the solution obtained through Algorithm 1 may not satisfy the $\epsilon$-KKT conditions of the original problem. Nonetheless, our analysis demonstrates that by setting $\tilde{\boldsymbol{w}}^k = \mathrm{prox}_{g,\gamma}(\boldsymbol{w}^k)$, our algorithm guarantees an $\epsilon$-KKT point of the original problem in at most $O(1/\epsilon^4)$ iterations.

## B Additional Experimental Details

### B.1 Source Code

The source code is available in the https://github.com/RufengXiao/ADMM-for-rank-based-loss.

## B.2 Experimental Environment

All algorithms are implemented in Python 3.8 and all the experiments are conducted on a Linux server with 256GB RAM and 96-core AMD EPYC 7402 2.8GHz CPU.

## B.3 Details for the PAVA

**Proposition 3** *Given two consecutive blocks $[m, n]$ and $[n + 1, p]$, if they are out-of-order, then we have*

$$\frac{\sum_{i=m}^{n} \sigma_i}{n - m + 1} < \frac{\sum_{i=n+1}^{p} \sigma_i}{p - n}.$$

*Proof:* Suppose on the contrary that $v_{[m,n]} > v_{[n+1,p]}$ but

$$\frac{\sum_{i=m}^{n} \sigma_i}{n - m + 1} \geq \frac{\sum_{i=n+1}^{p} \sigma_i}{p - n}.$$

We use the conventions that $h_{[a,b]}(z) = \sum_{i=a}^{b} \theta_i(z)$. For any $g \in \partial l(v_{[m,n]})$, we must have $g \geq 0$ because $l$ is monotonically increasing (see the beginning of Section 3.1). Therefore we have

$$\frac{\sum_{i=m}^{n} \sigma_i}{n - m + 1} g + \rho \left( v_{[m,n]} - \frac{\sum_{i=m}^{n} m_i}{n - m + 1} \right)$$

$$\geq \frac{\sum_{i=n+1}^{p} \sigma_i}{p - n} g + \rho \left( v_{[m,n]} - \frac{\sum_{i=m}^{n} m_i}{n - m + 1} \right)$$

$$\geq \frac{\sum_{i=n+1}^{p} \sigma_i}{p - n} g + \rho \left( v_{[m,n]} - \frac{\sum_{i=n+1}^{p} m_i}{p - n} \right),$$

where the first inequality is due to the non-negativity of $g$, and the second inequality is due to the fact that $m_i \leq m_{i+1}$ for $i = m, m + 1, \ldots, p$. Note also that

$$\frac{\sum_{i=m}^{n} \sigma_i}{n - m + 1} g + \rho \left( v_{[m,n]} - \frac{\sum_{i=m}^{n} m_i}{n - m + 1} \right) \in \frac{1}{n - m + 1} \partial h_{[m,n]}(v_{[m,n]})$$

and

$$\frac{\sum_{i=n+1}^{p} \sigma_i}{p - n} g + \rho \left( v_{[m,n]} - \frac{\sum_{i=n+1}^{p} m_i}{p - n} \right) \in \frac{1}{p - n} \partial h_{[n+1,p]}(v_{[m,n]})$$

We also have $0 \in \partial h_{[m,n]}(v_{[m,n]})$ from the optimality condition. This implies that there exists some $s \in \partial h_{[n+1,p]}(v_{[m,n]})$ such that $s \leq 0$.

Now if there exists another $s' \in \partial h_{[n+1,p]}(v_{[m,n]})$ such that $s' > 0$, then we have $0 \in \partial h_{[n+1,p]}(v_{[m,n]})$, which contradicts with the uniqueness of the optimal point $v_{[n+1,p]}$ to $h_{[n+1,p]}$ (due to its strongly convexity). So we must have $s \leq 0$ for all $s \in \partial h_{[n+1,p]}(v_{[m,n]})$. Then we must have $v_{[m,n]} \leq v_{[n+1,p]}$ from the optimality of $v_{[n+1,p]}$ to $h_{[n+1,p]}$ and convexity of $h_{[n+1,p]}$. However, this contradicts the fact that $[m, n]$ and $[n + 1, p]$ are *out-of-order*. ∎

Proposition 3 presents an additional opportunity to accelerate the PAVA, particularly for the top-$k$ loss and AoRR framework. Consider the case of top-$k$ loss, where $\sigma_i = 1$ if $i = k$ and $\sigma_i = 0$ otherwise. In this scenario, the condition for merging blocks is satisfied only when $i = k$. Therefore, we only need to examine the block containing $k$, its preceding block, and its subsequent block. Consequently, the time complexity of searching for *out-of-order* blocks in line 5 of Algorithm 2 is reduced to $O(1)$.

By leveraging this insight, we enhance the efficiency of our algorithm, specifically for top-$k$ loss. An analogous acceleration can also be applied to the AoRR loss. This optimization dramatically reduces computational complexity, thereby facilitating faster execution of the PAVA for both top-$k$ loss and AoRR loss.

In our implementation of the PAVA, we use either the bisection method or Newton's method to find a minimizer of the convex function $h_{[m,n]}$. Assuming the maximum number of iterations for these methods as $T$, the computation of the minimizer $v_{[m,n]}$ exhibits a time complexity of $O(T)$. In the PAVA, as we merge *out-of-order* blocks, the size of the index set $J$ decreases by at least one.

Initially, $J$ comprises $n$ blocks. Hence, the minimizer needs to be computed no more than $O(n)$ times throughout the algorithm. Furthermore, before each merge, we perform up to $O(n)$ comparisons. However, this complexity is reduced to $O(1)$ for top-$k$ loss and AoRR loss due to the structures of these losses. Overall, the time complexity of our PAVA is $O(n^2 + nT)$. Notably, for top-$k$ loss and AoRR loss, the time complexity is reduced to $O(n + nT)$.

Moreover, in the empirical human risk minimization with CPT-weight in (4), $\theta_i(z_i)$ is nonconvex with regard to $z_i$ because $\sigma_i$ is dependent on the value of $z_i$. Specifically, $\theta_i(z_i)$ takes the form of a two-piece function for $z_i \in (-\infty, B]$ and $(B, \infty)$, where $B$ represents a certain threshold. However, despite its nonconvexity, $\theta_i(z_i)$ remains convex within each piece. Exploiting this property, we can determine the minimizer of such a function by comparing the minimizers of the two separate pieces. Considering this observation, the overall time complexity of the PAVA for the empirical human risk minimization is still $O(n^2 + nT)$.

## B.4 Datasets Description

Datasets for our experiments are generated in two ways.

**Synthetic data.** We construct synthetic datasets where the data matrix $X$ and label vector $y$ are generated artificially. We utilize the `datasets.make_classification()` function from the Python package `scikit-learn` [38] to generate two-class classification problem datasets of various dimensions.

**Real data.** In the case of real data, the data matrix $X$ and label vector $y$ are derived from existing datasets. The 'SVMguide' [26] dataset, frequently used in support vector machines, is included in our experiments. We also employ 'AD' [29], which comprises potential advertisements for Internet pages, and 'Monks' [50], the dataset based on the MONK's problem that served as the first international comparison of learning algorithms. The 'Splice' dataset from the UCI [18] is used for the task of recognizing DNA sequences as exons or introns. We additionally include 'Australian', 'Phoneme', and 'Titanic' dataset from [17]. Lastly, the 'UTKFace' dataset [53] is used to predict gender based on facial images. Detailed statistics for each dataset are presented in Table 5.

Table 5: Statistical details of eight real datasets.

| Datasets | # Classes | # Samples | # Features | Class Ratio |
|---|---|---|---|---|
| AD | 2 | 2,359 | 1,558 | 0.20 |
| SVMguide | 2 | 3,089 | 4 | 1.84 |
| Monks | 2 | 432 | 6 | 1.12 |
| Australian | 2 | 690 | 14 | 0.80 |
| Phoneme | 2 | 5,404 | 5 | 0.41 |
| Titanic | 2 | 2,201 | 3 | 0.48 |
| Splice | 2 | 3,190 | 60 | 0.93 |
| UTKFace | 2 | 9,778 | 136 | 1.24 |

## B.5 Details of Our Algorithm Setting

In the experiments, we set the maximum iteration limit in our algorithm to 300. We utilize FISTA to solve the $w$-subproblem in our algorithm and L-BFGS for the smoothed $w$-subproblem. For varying frameworks, we adopt different choices of $\{\rho_k\}$, which varies when the iteration number $k$ increase, in Algorithm 1. To enable the replication of our experiments, we provide a suggested selection for $\{\rho_k\}$. The specifics regarding these choices for $\{\rho_k\}$ are detailed in Table 6. We set the $\gamma$ in Section 4 to $\gamma^k = \max\{10^{-5} \times 0.9^k, 10^{-9}\}$.

## B.6 Details of Experiments Setting

**Spectral Risk Measures.** Our objective is to demonstrate the versatility of our algorithm and its ability to converge to a globally optimal solution in the convex problem. Consequently, we utilize two real binary classification problem datasets and several synthetic datasets to highlight the advantages of our algorithm. We compare our method with the algorithms in [36]. We adapt their implementation

Table 6: The choices for $\{\rho_k\}$

| Framework | $\rho_0$ | $\rho_k$ |
|---|---|---|
| SRM | $10^{-5}$ | $1.2^k \rho_0$ |
| AoRR | $2 \times 10^{-7}$ | $5^{\lfloor (k-7)/3 \rfloor} \rho_0$ |
| EHRM | $10^{-4}$ | if $\|\boldsymbol{z}_k - D_k \boldsymbol{w}_k\|_2 > 10^{-2}$ then $\rho_k = 1.02\rho_{k-1}$ else $\rho_k = 1.07\rho_{k-1}$ |

by replacing the gradient of the $\ell_2$ norm with the soft threshold operator to incorporate the $\ell_1$ regularization. For simplicity, we set the regularization parameter $\mu$ as 0.01 and $q$ as 0.8. The datasets are divided into 60% for training and 40% for testing. Both SGD and LSVRG are run for 2000 epochs and the batch size of SGD is set to 64, and the epoch lengths are set to 100 as recommended in [36]. We give an example of how we choose the learning rate. For comparative purposes, we have chosen the learning rate that yields the lowest objective value in a single randomized experiment among five different learning rates from $\{10^{-2}, 10^{-3}, 10^{-4}, 1/N_{\text{samples}}, 1/(N_{\text{samples}} \times N_{\text{features}})\}$. We illustrate this approach using the AD dataset under the SRM superquantile framework as an example. The results are compiled in Table 7. The learning rate corresponding to the bold values, indicating the minimal objective value, is subsequently chosen.

Table 7: Objective value in different learning rates for SGD and LSVRG.

| | Logistic Loss | | | Hinge Loss | | |
|---|---|---|---|---|---|---|
| Learning Rate | LSVRG(NU) | LSVRG(U) | SGD | LSVRG(NU) | LSVRG(U) | SGD |
| $g(\boldsymbol{w}) = \frac{\mu}{2}\|\boldsymbol{w}\|_2^2$ | | | | | | |
| 0.01 | 0.49612 | 0.19169 | **0.16200** | 0.18016 | 0.22529 | 0.11009 |
| 0.001 | **0.15757** | **0.15753** | 0.16210 | **0.08600** | 0.09591 | **0.08702** |
| 0.0001 | 0.17936 | 0.17930 | 0.27038 | 0.09270 | 0.09261 | 0.11568 |
| $1/N_{\text{samples}}$ | 0.15761 | 0.15758 | 0.16755 | 0.08609 | **0.08940** | 0.08911 |
| $1/(N_{\text{samples}} \times N_{\text{features}})$ | 0.65849 | 0.65849 | 0.68518 | 0.85333 | 0.85333 | 0.96771 |
| $g(\boldsymbol{w}) = \frac{\mu}{2}\|\boldsymbol{w}\|_1$ | | | | | | |
| 0.01 | 0.34731 | 0.37191 | **0.27573** | 0.56240 | 0.45764 | 0.17956 |
| 0.001 | **0.26991** | **0.26986** | 0.28041 | 0.16954 | 0.16959 | **0.15089** |
| 0.0001 | 0.30385 | 0.30385 | 0.38138 | **0.15621** | **0.15471** | 0.18250 |
| $1/N_{\text{samples}}$ | 0.27084 | 0.27078 | 0.28763 | 0.16805 | 0.15828 | 0.15198 |
| $1/(N_{\text{samples}} \times N_{\text{features}})$ | 0.66741 | 0.66743 | 0.68723 | 0.87358 | 0.87358 | 0.97219 |

**Average of Ranked Range Aggregate Loss.** We employ the same real datasets as those used in [27]. Consistent with their experimental setup, we randomly split each dataset into a 50% training set, a 25% validation set, and a 25% test set. As [27] only considered $\ell_2$ regularization with $\mu = 10^{-4}$, our experiments in this section also utilize the AoRR aggregate loss with $\ell_2$ regularization. The choice of hyper-parameters $k$ and $m$ and the selection of inner and outer epochs for each dataset follow [27].Table 8 presents the hyper-parameters for individual logistic loss, while Table 9 displays those for individual hinge loss.

**Empirical Human Risk Minimization.** We set $\gamma = 0.61$ and $\delta = 0.69$ in (5) based on the recommendations in [45]. We set $B = \log(1 + \exp(-5))$ in (4). For simplicity, we only use $\ell_2$ regularization. We employ the 'UTKFace' dataset [53], also used in [31], for gender prediction. The experiments are performed five times with different random seeds. In this experimental segment, we exclusively use the logistic loss as the individual loss since hinge loss has most of the zeros so that it is not easy to determine the corresponding $B$ in (4) which means that its distribution is not suitable for this framework. The cumulative distribution function of loss $F$ is derived from synthetic data, and $F(B)$ is approximately equal to 0.05. The datasets are divided into 60% for training and 40% for testing. Other settings are similar to those in [31]. In accordance with [31], we divide the population into two groups based on race (white $G_1$ and other race $G_2$). However, for simplicity, we only obtain the parameters $w$ for gender prediction by minimizing the problem (1) without using a neural network. The parameter settings mirror those in Section 5.1. We use the same method to get the learning rate, and the learning rate for all three algorithms is set to $1/(N_{\text{samples}} \times N_{\text{features}})$.

Let TPR denote the true positive rate, FPR the false positive rate, and FNR the false negative rate. Additionally, let $X \in \mathcal{X}$ represent the dataset and $Y \in \{0, 1\}$ the label. We utilize the same fairness metrics suggested by [5], as followed in [31]. Considering the privileged group $G_1 \subseteq X$ and the unprivileged group $G_2 \subseteq X$, the definitions of the following metrics are from [31]:

- Statistical Parity Difference (SPD): $P(Y = 1 \mid X \in G_2) - P(Y = 1 \mid X \in G_1)$.

- Disparate Impact (DI): $\frac{P(Y=1|X \in G_2)}{P(Y=1|X \in G_1)}$.

- Equal Opportunity Difference (EOD): $\text{TPR}(G_2) - \text{TPR}(G_1)$.

- Average Odds Difference (AOD): $\frac{1}{2}(\text{FPR}(G_2) - \text{FPR}(G_1) + (\text{TPR}(G_2) - \text{TPR}(G_1)))$.

- Theil Index (TI): $\frac{1}{n}\sum_{i=1}^{n} \frac{b_i}{\mu} \ln\left(\frac{b_i}{\mu}\right)$ where $b_i = \widehat{Y}_i - Y_i + 1$ and $\mu = \frac{1}{n}\sum_{i=1}^{n} b_i$. Here, $\widehat{Y}_i$ is the prediction of $X_i$ and $n$ represents the number of samples.

- False Negative Rate Difference (FNRD): $\text{FNR}(G_2) - \text{FNR}(G_1)$.

Table 8: AoRR hyper-parameters on real datasets for individual logistic loss.

| Datasets | $k$ | $m$ | # Outer epochs | # Inner epochs | Learning rate |
|---|---|---|---|---|---|
| Monk | 70 | 20 | 5 | 2000 | 0.01 |
| Australian | 80 | 3 | 10 | 1000 | 0.01 |
| Phoneme | 1400 | 100 | 10 | 1000 | 0.01 |
| Titanic | 500 | 10 | 10 | 1000 | 0.01 |
| Splice | 450 | 50 | 10 | 1000 | 0.01 |

Table 9: AoRR hyper-parameters on real datasets for individual hinge loss.

| Datasets | $k$ | $m$ | # Outer epochs | # Inner epochs | Learning rate |
|---|---|---|---|---|---|
| Monk | 70 | 45 | 5 | 1000 | 0.01 |
| Australian | 80 | 3 | 5 | 1000 | 0.01 |
| Phoneme | 1400 | 410 | 10 | 500 | 0.01 |
| Titanic | 500 | 10 | 5 | 500 | 0.01 |
| Splice | 450 | 50 | 10 | 1000 | 0.01 |

## C   Additional Experimental Results

In this section, synthetic datasets labeled 1, 2, 3, and 4 are utilized. Detailed descriptions of datasets 1, 2, 3, and 4 can be found in Table 10.

Table 10: Details of the synthetic datasets.

| Datasets | num_sample | num_feature | Datasets | num_sample | num_feature |
|---|---|---|---|---|---|
| 1 | 1000 | 500 | 3 | 5000 | 1000 |
| 2 | 2000 | 1000 | 4 | 10000 | 1000 |

### C.1   Experiments of Synthetic Datasets with ERM

The experimental settings implemented in this section are consistent with those discussed in Section 5.1. The datasets are replaced with the synthetic datasets. We use the same method to get the learning rate for other algorithms. Tables 11, 12, and 13 enumerate the mean and standard deviation of the results, primarily focusing on the objective value and test accuracy. 'sADMM' stands for the ADMM applied to the smoothed problem. Tables 11, 12, and 13 show that under most scenarios in the ERM framework, our ADMM algorithm exhibits superior performance in terms of objective

values when compared to existing methods. This superiority is particularly evident in instances of hinge loss, with the algorithm demonstrating a test accuracy that is analogous to other methods.

Figure 2 graphically demonstrates the correlation between time and sub-optimality for each algorithm within the superquantile framework and hinge loss, which represents a convex problem whose global optimum is achievable. The figure indicates that ADMM displays a more rapid convergence towards the minimum relative to other algorithms. While sADMM does not match the performance of ADMM, it does achieve a commendable rate of convergence when compared to other algorithms.

Table 11: Results in synthetic datasets with SRM ERM framework and $\ell_2$ regularization. 'objval' denotes the objective value of problem (1).

| Datasets | | Logistic Loss | | | | Hinge Loss | | | |
|---|---|---|---|---|---|---|---|---|---|
| | | ADMM | LSVRG(NU) | LSVRG(U) | SGD | ADMM | LSVRG(NU) | LSVRG(U) | SGD |
| 1 | objval | 0.0923 (0.0057) | 0.0923 (0.0057) | 0.0923 (0.0057) | 0.0923 (0.0057) | 0.0088 (0.0011) | 0.0095 (0.0013) | 0.0096 (0.0012) | 0.0091 (0.0012) |
| | Accuracy | 0.7875 (0.0113) | 0.7875 (0.0113) | 0.7875 (0.0113) | 0.7875 (0.0113) | 0.7590 (0.0185) | 0.7770 (0.0143) | 0.7765 (0.0153) | 0.7760 (0.0171) |
| 2 | objval | 0.0800 (0.0012) | 0.0800 (0.0012) | 0.0800 (0.0012) | 0.08 (0.0012) | 0.0068 (0.0002) | 0.0075 (0.0004) | 0.0075 (0.0003) | 0.0071 (0.0002) |
| | Accuracy | 0.8483 (0.0115) | 0.8485 (0.0111) | 0.8485 (0.0111) | 0.8485 (0.0111) | 0.8193 (0.0079) | 0.8330 (0.0072) | 0.8325 (0.0068) | 0.8303 (0.0060) |
| 3 | objval | 0.1704 (0.0037) | 0.1704 (0.0037) | 0.1704 (0.0037) | 0.1704 (0.0037) | 0.0732 (0.0045) | 0.0807 (0.0047) | 0.0809 (0.0044) | 0.0756 (0.0046) |
| | Accuracy | 0.8347 (0.0039) | 0.8347 (0.0039) | 0.8347 (0.0039) | 0.8341 (0.0035) | 0.7999 (0.0091) | 0.8138 (0.0075) | 0.8135 (0.0053) | 0.8091 (0.0090) |
| 4 | objval | 0.1395 (0.0014) | 0.1395 (0.0014) | 0.1395 (0.0014) | 0.1395 (0.0014) | 0.0664 (0.0020) | 0.0716 (0.0021) | 0.0716 (0.0021) | 0.0667 (0.0020) |
| | Accuracy | 0.9400 (0.0014) | 0.9401 (0.0014) | 0.9401 (0.0014) | 0.9402 (0.0014) | 0.9250 (0.0046) | 0.9330 (0.0029) | 0.9330 (0.0028) | 0.9261 (0.0049) |

Table 12: Results in synthetic datasets with SRM ERM framework, $\ell_1$ regularization, and logistic loss. 'objval' denotes the objective value of problem (1).

| Datasets | | ADMM | sADMM | LSVRG(NU) | LSVRG(U) | SGD |
|---|---|---|---|---|---|---|
| 1 | objval | 0.1829 (0.012) | 0.1829 (0.012) | 0.1829 (0.012) | 0.1829 (0.012) | 0.1845 (0.0118) |
| | Accuracy | 0.8490 (0.0076) | 0.8495 (0.0087) | 0.8490 (0.0076) | 0.8495 (0.008) | 0.8500 (0.0079) |
| 2 | objval | 0.1693 (0.0037) | 0.1693 (0.0037) | 0.1693 (0.0037) | 0.1693 (0.0037) | 0.1729 (0.0037) |
| | Accuracy | 0.9190 (0.0061) | 0.9193 (0.0061) | 0.9190 (0.0061) | 0.9190 (0.0061) | 0.9193 (0.0056) |
| 3 | objval | 0.2584 (0.0049) | 0.2584 (0.0049) | 0.2585 (0.0049) | 0.2585 (0.0049) | 0.2596 (0.0049) |
| | Accuracy | 0.9005 (0.0053) | 0.9005 (0.0053) | 0.9005 (0.0053) | 0.9005 (0.0053) | 0.9005 (0.0051) |
| 4 | objval | 0.1475 (0.0037) | 0.1476 (0.0037) | 0.1479 (0.0037) | 0.1479 (0.0037) | 0.1495 (0.0037) |
| | Accuracy | 0.9599 (0.0026) | 0.9599 (0.0027) | 0.9599 (0.0027) | 0.9599 (0.0027) | 0.9599 (0.0025) |

Table 13: Results in synthetic datasets with SRM ERM framework, $\ell_1$ regularization, and hinge loss. 'objval' denotes the objective value of problem (1).

| Datasets | | ADMM | sADMM | LSVRG(NU) | LSVRG(U) | SGD |
|---|---|---|---|---|---|---|
| 1 | objval | 0.0777 (0.0075) | 0.0782 (0.0074) | 0.0856 (0.0087) | 0.0856 (0.0088) | 0.0802 (0.0076) |
| | Accuracy | 0.8095 (0.0200) | 0.8085 (0.0181) | 0.8000 (0.0181) | 0.7995 (0.0181) | 0.8045 (0.0192) |
| 2 | objval | 0.0786 (0.0023) | 0.0790 (0.0023) | 0.0877 (0.0022) | 0.0876 (0.0023) | 0.0836 (0.0023) |
| | Accuracy | 0.8880 (0.0143) | 0.8895 (0.0148) | 0.8823 (0.0121) | 0.8833 (0.0105) | 0.8848 (0.0130) |
| 3 | objval | 0.1983 (0.0058) | 0.1985 (0.0058) | 0.2038 (0.0057) | 0.2037 (0.0058) | 0.2006 (0.0058) |
| | Accuracy | 0.8783 (0.0056) | 0.8776 (0.0057) | 0.8770 (0.0061) | 0.8780 (0.0057) | 0.8783 (0.0048) |
| 4 | objval | 0.1192 (0.0031) | 0.1193 (0.0031) | 0.1229 (0.0042) | 0.1221 (0.0027) | 0.1206 (0.0030) |
| | Accuracy | 0.9563 (0.0018) | 0.9565 (0.0015) | 0.9564 (0.0020) | 0.9564 (0.0021) | 0.9561 (0.0019) |

## C.2 Experiments of Synthetic Datasets with AoRR

In the application of AoRR to synthetic data, we have configured the parameters $k$ and $m$ to be 0.8 and 0.2 times the size of the training set, respectively. In other words, $k = \lfloor 0.8N_{\text{train}} \rfloor$ and $m = \lceil 0.2N_{\text{train}} \rceil$. We maintain the learning rate in the DCA at 0.01, and set the outer epoch at 10, and the inner epoch at 2000. The parameters for SGD and LSVRG are identical to those employed in the preceding experiment (the learning rate is chosen from the same method). Tables 14 and 15

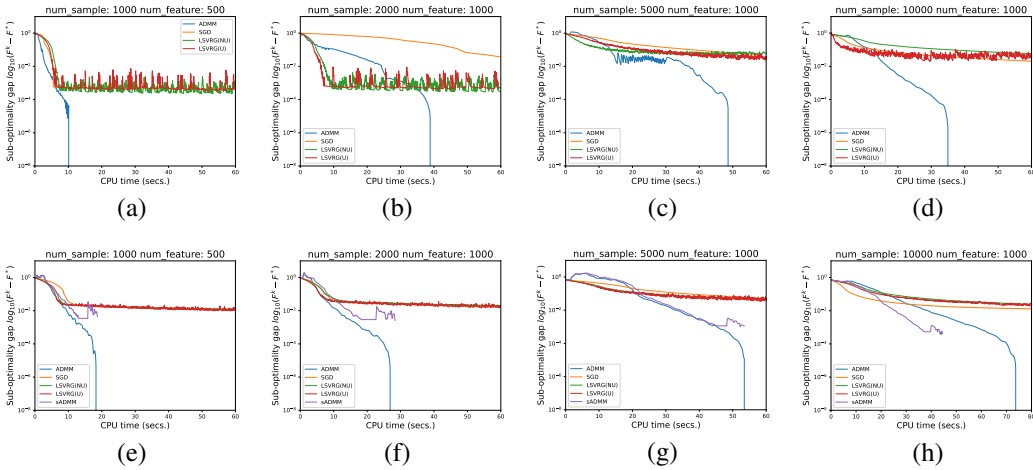

Figure 2: Time vs. Sub-optimality gap in synthetic datasets with superquantile framework. (a-d) for $\ell_2$ regularization, and (e-f) for $\ell_1$ regularization. Sub-optimality is defined as $F^k - F^*$, where $F^k$ represents the objective function value at the $k$-th iteration or epoch and $F^*$ denotes the minimum value obtained by all algorithms. Plots are truncated when $F^k - F^* < 10^{-8}$.

enumerate the mean and standard deviation of results for the objective value, accuracy, and time. A careful examination of Tables 14 and 15 suggests that our algorithm, in most scenarios, achieves lower objective values and comparable accuracy in a shorter time span relative to existing methods.

Table 14: Comparison with AoRR framework, $k = \lfloor 0.8 \times N_{\text{train}} \rfloor$, $m = \lceil 0.2 \times N_{\text{train}} \rceil$ and logistic loss. 'objval' denotes the objective value of problem (1).

| Datasets | | ADMM | DCA | LSVRG(NU) | LSVRG(U) | SGD |
|---|---|---|---|---|---|---|
| 1 | objval | $1.75 \times 10^{-3}$ ($9.52 \times 10^{-5}$) | $7.12 \times 10^{-1}$ ($3.94 \times 10^{-2}$) | $2.10 \times 10^{-3}$ ($1.56 \times 10^{-4}$) | $2.14 \times 10^{-3}$ ($1.17 \times 10^{-4}$) | $4.65 \times 10^{-3}$ ($3.32 \times 10^{-4}$) |
| | Accuracy | 0.7641 (0.0429) | 0.7729 (0.0325) | 0.7514 (0.0329) | 0.7394 (0.0151) | 0.7737 (0.0312) |
| | Time | 11.43 (0.87) | 125.7 (21.88) | 110.24 (8.13) | 94.86 (7.58) | 285.43 (1.17) |
| 2 | objval | $1.41 \times 10^{-3}$ ($2.99 \times 10^{-5}$) | $7.22 \times 10^{-1}$ ($2.73 \times 10^{-2}$) | $1.71 \times 10^{-3}$ ($3.85 \times 10^{-5}$) | $1.81 \times 10^{-3}$ ($7.96 \times 10^{-5}$) | $2.21 \times 10^{-3}$ ($5.86 \times 10^{-5}$) |
| | Accuracy | 0.8283 (0.0073) | 0.8315 (0.0159) | 0.8248 (0.0158) | 0.808 (0.0094) | 0.8427 (0.0139) |
| | Time | 19.33 (1.98) | 107.27 (20.81) | 124.14 (12.98) | 104.22 (4.69) | 399.86 (11.15) |
| 3 | objval | $2.75 \times 10^{-3}$ ($6.88 \times 10^{-5}$) | $7.49 \times 10^{-1}$ ($1.22 \times 10^{-2}$) | $3.00 \times 10^{-3}$ ($9.22 \times 10^{-5}$) | $3.07 \times 10^{-3}$ ($1.14 \times 10^{-4}$) | $3.33 \times 10^{-3}$ ($1.09 \times 10^{-4}$) |
| | Accuracy | 0.8149 (0.0109) | 0.7936 (0.0112) | 0.8083 (0.0095) | 0.8075 (0.012) | 0.8222 (0.0058) |
| | Time | 56.95 (2.07) | 127.18 (19.86) | 161 (14.01) | 149.78 (8.62) | 746.75 (72.28) |
| 4 | objval | $3.24 \times 10^{-3}$ ($3.13 \times 10^{-5}$) | $7.43 \times 10^{-1}$ ($2.30 \times 10^{-2}$) | $3.23 \times 10^{-3}$ ($3.05 \times 10^{-5}$) | $3.27 \times 10^{-3}$ ($3.99 \times 10^{-5}$) | $3.20 \times 10^{-3}$ ($3.39 \times 10^{-5}$) |
| | Accuracy | 0.9417 (0.0039) | 0.9302 (0.0045) | 0.9325 (0.0056) | 0.9273 (0.0018) | 0.9385 (0.0044) |
| | Time | 222.35 (19.30) | 154.46 (43.08) | 222.37 (62.44) | 205.76 (64.40) | 525.84 (237.37) |

### C.3 Experiments about Figure 1

We increased the sample size to further observe the performance of our algorithm and the experimental results are shown in Figure 3. It can be seen that both ADMM and sADMM exhibit relatively fast convergence compared to other algorithms. An interesting phenomenon is that in the case of a large amount of data, random algorithms are able to achieve decent results in the early stages of optimization. The underlying reason for this phenomenon may be that existing methods use mini-batch samples that are independent of the sample size to update model parameters, thereby reducing the computational cost per iteration. This allows existing methods to update parameters more frequently, leading to faster convergence in the early stages of optimization. In contrast, the proposed algorithm uses the entire batch of samples in each iteration, resulting in slower iteration speeds. This leads to suboptimal solutions in the early stages compared to existing methods. In Appendix B.3, we provide a detailed explanation of the time complexity of the PAVA, which is $O(n + nT)$ for top-k loss and AoRR loss, where $T$ represents the maximum number of iterations when solving each PAVA subproblem, and $n$ represents the sample size. Therefore, with an increase in

Table 15: Comparison with AoRR framework, $k = \lfloor 0.8 \times N_{\text{train}} \rfloor$, $m = \lceil 0.2 \times N_{\text{train}} \rceil$ and hinge loss. 'objval' denotes the objective value of problem (1).

| Datasets | | ADMM | DCA | LSVRG(NU) | LSVRG(U) | SGD |
|---|---|---|---|---|---|---|
| | objval | $2.97 \times 10^{-5}$ ($1.92 \times 10^{-6}$) | $9.99 \times 10^{-1}$ ($2.66 \times 10^{-2}$) | $1.71 \times 10^{-3}$ ($1.49 \times 10^{-4}$) | $1.72 \times 10^{-3}$ ($1.38 \times 10^{-4}$) | $3.68 \times 10^{-5}$ ($2.34 \times 10^{-6}$) |
| 1 | Accuracy | 0.7700 (0.0371) | 0.7750 (0.0409) | 0.7600 (0.0427) | 0.7620 (0.0415) | 0.7680 (0.0315) |
| | Time | 4.37 (0.66) | 137.22 (28.90) | 116.39 (3.64) | 114.37 (2.93) | 286.19 (1.35) |
| | objval | $2.12 \times 10^{-5}$ ($6.51 \times 10^{-7}$) | $1.00 \times 10^{0}$ ($2.66 \times 10^{-2}$) | $7.05 \times 10^{-4}$ ($5.90 \times 10^{-5}$) | $1.21 \times 10^{-3}$ ($3.85 \times 10^{-4}$) | $2.42 \times 10^{-5}$ ($5.20 \times 10^{-7}$) |
| 2 | Accuracy | 0.8320 (0.0093) | 0.8270 (0.0099) | 0.8420 (0.0113) | 0.6940 (0.0277) | 0.8380 (0.0093) |
| | Time | 5.54 (0.64) | 172.5 (24.82) | 104.79 (2.87) | 93.35 (3.00) | 352.67 (4.98) |
| | objval | $4.82 \times 10^{-5}$ ($1.49 \times 10^{-6}$) | $9.89 \times 10^{-1}$ ($1.90 \times 10^{-2}$) | $2.94 \times 10^{-4}$ ($9.49 \times 10^{-6}$) | $2.78 \times 10^{-4}$ ($4.43 \times 10^{-5}$) | $6.30 \times 10^{-5}$ ($2.01 \times 10^{-6}$) |
| 3 | Accuracy | 0.8130 (0.0113) | 0.8060 (0.0084) | 0.8100 (0.0101) | 0.8070 (0.0135) | 0.8180 (0.0066) |
| | Time | 15.36 (0.70) | 290.7 (26.15) | 112.68 (8.08) | 99.52 (4.88) | 540.62 (66.63) |
| | objval | $8.53 \times 10^{-4}$ ($1.59 \times 10^{-4}$) | $9.82 \times 10^{-1}$ ($2.51 \times 10^{-2}$) | $1.22 \times 10^{-4}$ ($3.17 \times 10^{-6}$) | $1.08 \times 10^{-4}$ ($3.85 \times 10^{-6}$) | $8.16 \times 10^{-5}$ ($1.00 \times 10^{-6}$) |
| 4 | Accuracy | 0.9430 (0.0053) | 0.9360 (0.0046) | 0.9310 (0.0049) | 0.9240 (0.0028) | 0.9350 (0.0048) |
| | Time | 31.42 (3.34) | 396.03 (31.66) | 129.22 (14.09) | 115.06 (8.46) | 633.69 (82.42) |

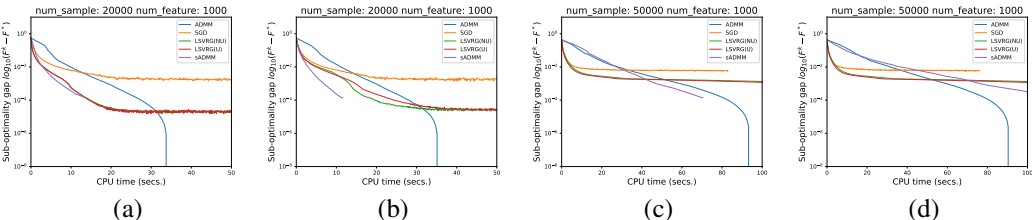

(a) (b) (c) (d)

Figure 3: Time vs. Sub-optimality gap in synthetic dataset with ERM framework and $\ell_1$ regularization. The datasets with the same number of samples are generated by different random number seeds. Sub-optimality is defined as $F^k - F^*$, where $F^k$ represents the objective function value at the $k$-th iteration or epoch and $F^*$ denotes the minimum value obtained by all algorithms. Plots are truncated when $F^k - F^* < 10^{-8}$.

sample size, the time required for our proposed algorithm will also increase. Nevertheless, compared to existing algorithms, we are still able to achieve higher accuracy within a reasonable time frame.

