(\hat{z}_{[i]}) = \begin{cases} \omega_-(\frac{i}{n}) - \omega_-(\frac{i-1}{n}), & \hat{z}_{[i]} \leq B, \\ \omega_+(\frac{n-i+1}{n}) - \omega_+(\frac{n-i}{n}), & \hat{z}_{[i]} > B, \end{cases} \tag{4}$$

where

$$\omega_+(p) = \frac{p^\gamma}{(p^\gamma + (1-p)^\gamma)^{1/\gamma}}, \quad \omega_-(p) = \frac{p^\delta}{(p^\delta + (1-p)^\delta)^{1/\delta}}, \tag{5}$$

with $\gamma$ and $\delta$ as hyperparameters.

**Ranked-Range Loss**  The average of ranked-range aggregate (AoRR) loss follows the same structure as (2), where

$$\sigma_i = \begin{cases} \frac{1}{k-m}, & i \in \{m+1, \ldots, k\}, \\ 0, & \text{otherwise}, \end{cases} \tag{6}$$

with $1 \leq m < k \leq n$. The ranked-range loss effectively handles outliers, ensuring the robustness of the model against anomalous observations in the dataset [27]. It is clear that AoRR includes the average loss, the maximum loss, and the average top-$k$ loss, and the median loss [34]. [27] utilized the difference-of-convex algorithm (DCA) [40] to solve the AoRR aggregate loss minimization problem, which can be expressed as the difference of two convex problems.

## 2.2 Weakly Convex Function and Moreau Envelope

A function $g : \mathbb{R}^d \to \mathbb{R}$ is said to be $c$-weakly convex for some $c > 0$ if the function $g + \frac{c}{2}\|\cdot\|^2$ is convex. The class of weakly convex functions is extensive, encompassing all convex functions as well as smooth functions with Lipschitz-continuous gradients [37]. In our framework, we consider weakly convex functions as regularizers. It is worth noting that weakly convex functions constitute a rich class of regularizers. Convex $L^p$ norms with $p \geq 1$ and nonconvex penalties such as the Minimax Concave Penalty (MCP) [52] and the Smoothly Clipped Absolute Deviation (SCAD) [20] are examples of weakly convex functions [7].

Next, we define the Moreau envelope of $c$-weakly convex function $g(\boldsymbol{w})$, with proximal parameter $0 < \gamma < \frac{1}{c}$:

$$M_{g,\gamma}(\boldsymbol{w}) = \min_{\boldsymbol{x}} \left\{ g(\boldsymbol{x}) + \frac{1}{2\gamma} \|\boldsymbol{x} - \boldsymbol{w}\|^2 \right\}. \tag{7}$$

The proximal operator of $g$ with parameter $\gamma$ is given by

$$\operatorname{prox}_{g,\gamma}(\boldsymbol{w}) = \arg\min_{\boldsymbol{x}} \left\{ g(\boldsymbol{x}) + \frac{1}{2\gamma} \|\boldsymbol{x} - \boldsymbol{w}\|^2 \right\}.$$

We emphasize that $\operatorname{prox}_{g,\gamma}(\cdot)$ is a single-valued mapping, and $M_{g,\gamma}(\boldsymbol{w})$ is well-defined since the objective function in (7) is strongly convex for $\gamma \in \left(0, c^{-1}\right)$ [7].

The Moreau envelope is commonly employed to smooth weakly convex functions. From [7] we have

$$\nabla M_{g,\gamma}(\boldsymbol{w}) = \gamma^{-1} \left(\boldsymbol{w} - \operatorname{prox}_{g,\gamma}(\boldsymbol{w})\right) \in \partial g \left(\operatorname{prox}_{g,\gamma}(\boldsymbol{w})\right). \tag{8}$$

Here $\partial$ represents Clarke generalized gradient [10]. Recall that for locally Lipschitz continuous function $f : \mathbb{R}^d \to \mathbb{R}$, its Clarke generalized gradient is donated by $\partial f(\boldsymbol{x})$. We say $\boldsymbol{x}$ is stationary for $f$ if $0 \in \partial f(\boldsymbol{x})$. For convex functions, Clarke generalized gradient coincides with subgradient in the sense of convex analysis. Under assumptions in Section 3.3, both $\Omega$ and $g$ are locally Lipschitz continuous [12, 47]. Thus the stationary point of $\Omega$ and $g$ can be characterized by Clarke generalized gradient, so is $