# OpenReview forum: "A Unified Framework for Rank-based Loss Minimization"
_NeurIPS.cc/2023/Conference — NeurIPS 2023 poster_

### Official Review · Reviewer_kXbu · 2023-07-05

**Soundness:** 3 good
**Presentation:** 2 fair
**Contribution:** 2 fair
**Rating:** 4
**Confidence:** 2

**Summary:**

This paper presents rank-based loss as a popular replacement for empirical loss. The work develops how
optimization of rank-based loss can be done by a proximal alternating direction method of multipliers. The authors also demonstrate the algorithm's features in terms of convergence under certain conditions. Experimentation which includes synthetic and real datasets is also added in the paper with some numerical simulations showing how the framework behaves. It supports the theoretical results.

**Strengths:**

The paper is mathematically strong

**Weaknesses:**

The contribution doesn't seem significant enough
The applicability of the result is not clearly shown


**Questions:**

I would like the authors to compare the method with similar-purpose frameworks, even if not using rank-based loss, and see their differences in performance.

I would appreciate the inclusion of the applicability of the result to a real machine-learning problem.



**Limitations:**

N.A.

---

> ### Author Rebuttal · Authors · 2023-08-05
>
> We thank reviewer kXbu for the constructive comments!
>
> **The applicability in machine learning**
>
> It is noteworthy that rank-based loss is a highly valuable and extensively researched concept in the field of machine learning. Commonly encountered variations of rank-based losses include spectral risk measures [1], empirical human risk [4], and the average of ranked-range aggregate loss [3]. These losses have been demonstrated to be significant and applicable in various scenarios, as expounded in the literature. Spectral risk measures, which include the well-known CVaR loss,  the maximum risk [5], and the average top-k risk [2], are widely used risk metrics in the field of machine learning and finance. Empirical human risk is an aggregate loss framework inspired by cumulative prospect theory [6], which considers fairness performance across different groups. The average of ranked-range aggregate loss is particularly effective in handling outliers, ensuring the robustness of the model against anomalous observations in the dataset.
>
> This paper considers a unified framework, equation (1), which encompasses all the aforementioned rank-based losses. Furthermore, we introduce a method to tackle problems within this unified framework. Our proposed algorithm exhibits comparable or even superior performance compared to algorithms specifically tailored for the three individual frameworks, as demonstrated in the experimental results.
>
> The versatility of our framework, along with the promising results obtained in experiments, highlights the potential of our approach in addressing various rank-based loss scenarios in machine learning applications.
>
> **Comparison with the existing method**
>
> Sorry that we may not understand your first question correctly.
> To our best knowledge, there does not exist any method that can address a unified framework akin to equation (1). Our proposed algorithm is the first one that is applicable to this unified framework. Nevertheless, we have compared different algorithms for solving (1) with different risk measures in the current version of our paper.
>
> In the experiments, we conducted comprehensive comparisons with algorithms used or proposed in prior work for the three application scenarios, i.e., LSVRG, SGD, and DCA . Our algorithm consistently demonstrated stability and superior performance in these experiments. Additionally, we examined the commonly used empirical risk minimization (ERM), which can be seen as not using rank-based loss, and presented the results in Figure 1 of Section 5.1 and Appendix C.1. In this setting, SGD refers to the standard stochastic gradient descent method, which is a commonly used optimization algorithm in machine learning.
>
> Once again, we emphasize that our algorithm offers a comprehensive and effective solution for various rank-based loss scenarios within the unified framework presented in equation (1). The experimental results consistently demonstrate the robustness and efficacy of our proposed algorithm.
>
> **The real machine-learning problem**
>
> In our study, we primarily focused on presenting the binary classification scenario to emphasize the effectiveness of our algorithm in solving problems within the equation (1) framework, with a particular focus on comparing the objective function values. However, we also included classification accuracy results to provide a comprehensive evaluation.
>
> In Tables 1 and 2, we displayed the classification accuracies of the spectral risk measure minimization model on the 'SVMguide' and 'AD' binary classification datasets (detailed sources and statistical information are provided in the appendix). The results show that our algorithm achieves comparable classification accuracy with existing methods.
> Additionally, in Table 3, we demonstrated the results of empirical human risk minimization on the 'UTKFace' dataset for the binary gender classification task. The results indicate that our algorithm outperforms existing methods in terms of test accuracy. Regarding fairness metrics, detailed explanations are available in Appendix B.5. For our analysis, we used race groups (white G1 and other race G2) to compare fairness metrics. The results reveal that our algorithm performs favorably on most fairness metrics compared to existing methods.
>
>
> **References**
>
> [1] Mehta, R., Roulet, V., Pillutla, K., Liu, L., & Harchaoui, Z. (2023, April). Stochastic Optimization for Spectral Risk Measures. In International Conference on Artificial Intelligence and Statistics (pp. 10112-10159). PMLR.
>
> [2] Fan, Y., Lyu, S., Ying, Y., and Hu, B. (2017). Learning with average top-k loss. Advances in neural information processing systems, 30.
>
> [3] Hu, S., Ying, Y., Lyu, S., et al. (2020). Learning by minimizing the sum of ranked range. Advances in Neural Information Processing Systems, 33:21013–21023.
>
> [4] Leqi, L., Prasad, A., and Ravikumar, P. K. (2019). On human-aligned risk minimization. Advances in Neural Information Processing Systems, 32.
>
> [5] Shalev-Shwartz, S. and Wexler, Y. (2016). Minimizing the maximal loss: How and why. In International Conference on Machine Learning, pages 793–801. PMLR.
>
> [6] Tversky, A. and Kahneman, D. (1992). Advances in prospect theory: Cumulative representation of uncertainty. Journal of Risk and uncertainty, 5:297–323.

---

> > ### Comment · Reviewer_kXbu · 2023-08-16
> > **acknowledgement of having read the authors' reply**
> >
> > I thank the authors for addressing all the points I indicated and providing a reply to them.

---

### Official Review · Reviewer_G6DJ · 2023-07-05

**Soundness:** 3 good
**Presentation:** 3 good
**Contribution:** 3 good
**Rating:** 7
**Confidence:** 3

**Summary:**

This submission focuses on efficient minimization of a group of loss functions called rank-based losses. It proposes to consider several related losses from the perspective of a genral unified framework with a regularizer. Then, focusing on the case of monotone increasing loss functions and wealy convex regularizers, it proposes an ADMM-based algorithm and shows its convergence rate under common assumptions. Furthermore, when the regularizer is non-smooth, it then extends the proposed algorithm with weaker assumptions and also shows its convergence rate. Sufficient numerical verification shows the satisfying empirical performance of the proposed algorithm compared with several existing methods.


**Strengths:**

Originality:
- The task provides a new perspective on an important problem and the proposed methods improving ADMM are novel.
- Authors clearly address where exactly the improvements from existing methods are.
- Related literature review is adequate and detailed.

Quality:
- The submission is technically rigid.
- Claims are well-supported by its clear presentation. Adequate theoretical and empirical results are presented.

Clarity:
- The submission is clearly written and easy to follow. It is very well organized and the story line expands naturally.

Significance:
- The results are important to the field. Others are very likely to use the results as a baseline method or build extensions upon it.



**Weaknesses:**

- It would be more pursuative to elaborate on potential practical limitations of the proposed method. For example, under what case the proposed method may not function efficiently.


**Questions:**

- Is equation (1) first proposed by this paper, or it has been somehow mentioned in related publications?
- In Figure 1 (h), it seems like existing methods can take advantage of more data at the beginning of optimization. How this trend changes when more samples are available? Is there some theoretical explainable available for the phenomenon?


**Limitations:**

No potential negative societal impact needs to be addressed.

---

> ### Author Rebuttal · Authors · 2023-08-05
>
> We thank reviewer G6DJ for the positive feedback and comments!
>
> **The potential practical limitations of the proposed method**
>
> To utilize our algorithm effectively, it is essential for the individual loss to exhibit monotonicity, as this allows us to employ the PAV algorithm to solve the $z$-subproblems. However, on the other hand, when dealing with a considerably large sample size, the computational efficiency of the PAV algorithm might be affected, potentially limiting the overall effectiveness of the entire algorithm. We will point out the limitations in our paper.
>
> **More information about equation (1)**
>
> In other papers, we have also observed similar equation representations. For example, in [1], the authors presented a similar equation as follows,
>
> \begin{equation}
> \mathcal{R}_\sigma(\boldsymbol{w})+\frac{\mu}{2}\\|\boldsymbol{w}\\|_2^2 \quad \text { for }  \\mathcal{R}\_\\sigma(\\boldsymbol{w}) = \sum\_i \sigma_i \boldsymbol{l}\_{[i]}(\boldsymbol{w})
> \end{equation}
>
> with $0 \leq \sigma_1 \leq \dots \sigma_n \leq 1, \sum_{i=1}^n \sigma_i=1$. We replace $\boldsymbol{l}\_{[i]}(w)$ by $\boldsymbol{l}\_{[i]}\left(- \boldsymbol{y} \odot (X\boldsymbol{w})\right)$ in order to obtain easily solvable subproblems of the ADMM algorithm. Moreover, in equation (1), we only require $\sigma_i\geq 0$, which includes a wider range of ranked-based losses, and allow $\sigma_i$ (in $\sigma_i(z_i)l(z_i)$) have different values depending on the values of $z_i$ being larger than or less than the reference point, which includes the human risk loss.
>
> We sincerely apologize for the typos in equation (1) which should be $\boldsymbol{l}:\mathbb{R}^n \to \mathbb{R}^n$ and $g: \mathbb{R}^d \to \mathbb{R}$.  Here $\boldsymbol{l}:\mathbb{R}^n \to \mathbb{R}^n$ represents a vector-valued mapping, where its  $i$-th element represents the individual loss for the $i$-th sample.
>
> **The explanation about Figure 1 (h)**
>
> This is a very interesting observation. Following your question, we varied the random seed to generate different datasets to eliminate the impact of randomness, and we still observed the same phenomenon across different datasets;
> moreover, we increased the sample size to explore trends, but the phenomenon remained unchanged.
> The corresponding experimental plots are included in the submitted PDF file.
>
>
> The potential reason for this phenomenon could be that existing methods update their model parameters using a mini-batch of the sample that is independent of the sample size, resulting in lower computational costs per iteration. This enables other algorithms to update parameters more frequently, leading to faster convergence at the beginning of optimization. In contrast, the proposed algorithm uses the full batch of the sample in each iteration, so each iteration is slower. This, in turn, leads to inferior solutions compared to existing methods at the beginning of optimization. In Appendix B.2, we provide a detailed explanation of the time complexity of the PAV algorithm, which is $O(n + nT)$ for top-k loss and AoRR loss, where $T$ represents the maximum number of iterations when solving each PAV subproblem and $n$ is the sample size. As a result, as the sample size increases, the time required by our proposed algorithm increases. Nevertheless, we can still achieve higher accuracy within a reasonable time frame compared to existing algorithms.
>
> In future research, we will aim to develop a variant of the proposed algorithm that uses a mini-batch of samples instead of using the full batch to achieve better initial solutions at the beginning of optimization.
>
> **Reference**
>
> [1] Mehta, R., Roulet, V., Pillutla, K., Liu, L., & Harchaoui, Z. (2023, April). Stochastic Optimization for Spectral Risk Measures. In International Conference on Artificial Intelligence and Statistics (pp. 10112-10159). PMLR.

---

> > ### Comment · Reviewer_G6DJ · 2023-08-16
> >
> > I thank authors for providing detailed replies for all my and other reviewers concerns.
> >
> > Authos clarified my concerns over the difference of similar equations to Eq(1) in related literatures. The consideration for faster update of existing methods in experiments is very thoughtful.

---

### Official Review · Reviewer_GkCz · 2023-07-06

**Soundness:** 3 good
**Presentation:** 3 good
**Contribution:** 3 good
**Rating:** 5
**Confidence:** 2

**Summary:**

This paper presents a new ADMM algorithm that focuses on three specific cases of rank-based losses. The algorithm's convergence is theoretically analyzed in the paper. Additionally, the authors conducted comprehensive experiments to compare the new algorithm with traditional approaches. The results indicate that the proposed algorithm outperforms traditional methods in terms of both efficiency and effectiveness.

**Strengths:**

1. The authors proposed a new algorithm that can address three specific cases of rank-based losses, whereas traditional algorithms like SGD, DCA, and LSVRG can only handle one. Additionally, the new algorithm permits the regularization term to be a weakly convex function.
2. The authors' theoretical analysis of the new algorithm's convergence is a significant contribution that has not been achieved in previous work.
3. To demonstrate the advantages of the new algorithm, the authors conducted comprehensive experiments. The results show that the new algorithm generally outperforms existing methods in terms of both efficiency and effectiveness. Overall, this study provides valuable insights into the development of more efficient and effective algorithms for rank-based losses.

**Weaknesses:**

Significant Issues (Detailed Explanation Required)
1.The representation of the loss vector function in line 23 is unclear. If function l maps an n-dimensional column vector to a real number, how a comparison of magnitude can be made later on?

2.As is widely known, due to the unknown underlying distribution of the data, we can only optimize the discrete form of risk expectation, which is the arithmetic mean of n observations. The theoretical basis for doing so is that the arithmetic mean converges in probability to the expectation. Therefore, could you please explain the theoretical basis for the discrete form of spectral risk that you employed in lines 78-79.

3.Could you please clarify how the conclusion stated in line 116 was derived? If it is a result obtained from citing other papers, please provide the source. If it is a result you have proven yourself, please provide a detailed proof.

4.In line 159, definitions for two consecutive blocks disorder are provided, but definitions for three or more consecutive block disorder are missing.

5.In line 6 of Algorithm 2, it is not possible to select the disordered blocks because definitions for three or more consecutive blocks disorder have not been provided.

6.In line 6 of Algorithm 2, you assume that the optimal solutions for  are equal to each other. Could you please a more detailed proof or a citation if the conclusion is derived from another paper?

7.In line 225, you stated that Assumptions 4 and 5 are weaker than Assumption 2. Please provide a detailed explanation for this claim.

8.Could you please provide specific information on the loss functions and regularizers used in the experiments in Sections 5.2 and 5.3?

Minor issue (needs improvement).

1.The "l" in line 23 and the "l" in line 71 have different meanings. It’s suggested to use distinct notations to indicate this difference.

2.In Equation (1), the independent variable of  is , where  is a d-dimensional column vector. However, the independent variable of  is written as an n-dimensional column vector in line 24.

3.Line 71 should use lowercase "d" instead of uppercase.

4.In line 85, please provide a brief explanation of the term "fairness".

5.There is a missing negative sign in line 127.

6.In line 154, "" should be in lowercase.

7.The explanations in lines 169-170 appear to contradict the explanations in lines 187-188.

8.To avoid confusion among readers, please use a consistent notation for the loss function "" in lines 71, 125, and 194.

9.Please provide an explanation for "dist" as used in line 249 at its initial occurrence in line 212.

10.The placement of "Dataset" in Tables 1 and 2 is inconsistent.

11.Please use the notation "new-ADMM" to distinguish the modified version of the traditional ADMM algorithm.

**Questions:**

Please see the above comments about  the Weakness.

---

> ### Author Rebuttal · Authors · 2023-08-05
>
> We thank reviewer GkCz for the feedback and comments!
>
> **The confusion in notations for the loss function**
>
> We apologize for the confusion. We should only use the definitions of $l$  and $\boldsymbol{l}$ as follows:
>
> 1. $l$: $\\mathbb{R}\\to\\mathbb{R}$: a  function that represents the loss for an individual sample.
> 2. $\\boldsymbol{l}$: $\\mathbb{R}^n\\to\\mathbb{R}^n$: a vector-valued function whose $i$-th element represents the individual loss for the $i$-th sample.
>
> We will correct all the places where the loss function is used incorrectly.
>
> **Theoretical basis for the discrete form of spectral risk**
>
> The discrete form of the spectral risk converges to the spectral risk of the population distribution, governed by Wasserstein distance. Please refer to [Proposition 1, 1].
>
> **Conclusion about Moreau Envelope**
>
> The proximal operator of a $c$-weakly convex function $g$ is $\\operatorname{prox}_{g,\\gamma}(w)=\\arg\\min _{x\\in\\mathbb{R}^d}\\left\\{g(x)+\\frac{1}{2 \\gamma}\\|x-w\\|^2\\right\\}$. According to the definition of the weakly-convex function in line 106 and given $0 < \\gamma < \\frac{1}{c}$, the function $\\mathbb{R}^d \\ni x\\to g(x)+\\frac{1}{2\\gamma} \\|x-w\\|^2\\in\\mathbb{R}$ is strongly convex for all $w\\in\\mathbb{R}^d$, thus its argmin is always a singleton.
>
> Furthermore, $ M_{g,\\gamma}(w)=\\min_x\\left\\{g(x)+\\frac{1}{2 \\gamma}\\|x-w\\|^2\\right\\}=g(\operatorname{prox}\_{g,\gamma}(w))+\\frac{1}{2\\gamma}\\|\\operatorname{prox}_{g,\\gamma}(w)-w\\|^2 $ is well-defined. A detailed proof can be found in [Proposition 3.1, 2].
>
> **Explanation of PAV algorithm**
>
> Sorry for missing the definition for three or more consecutive out-of-order blocks, which is similar with two consecutive out-of-order blocks. That is, if $v_{[s_k,s_{k+1}]} > v_{[s_{k+1} + 1,s_{k+2}]} > \cdots > v_{[s_{k+t}+1,s_{k+t+1}]}$, then $\\{[s_k,s_{k+1}],[s_{k+1} + 1,s_{k+2}], \cdots , [s_{k+t}+1,s_{k+t+1}]\\}$ are consecutive out-of-order blocks. For example, if $v_{[1,2]}<v_{[3,3]}<v_{[4,5]}>v_{[6,6]}>v_{[7,7]}<v_{[8,8]}$, then $\\{[4,5], [6,6], [7,7]\\}$ are a consecutive out-of-order blocks.
>
>
> Sorry that the statement in line 169-170 is unclear.
> We should point out that in  $\theta_i(z_i)=\sigma_i l(z_i)+\frac{\rho}{2}(z_i-m_i)^2$,  $\sigma_i$ is constant for the spectral risk loss or the ranked-range loss, but $\sigma_i$ is a function of $z_i$ in the human risk loss as it depends on the value of $z$ being larger than or less than the reference point $B$ (see  eq. (4)).
> So for cases that $\sigma_i$ is a constant, $\theta_i$ is a convex function, and the acceleration procedure via merging multiple consecutive out-of-order blocks is applicable. However, for human risk minimization, $\theta_i$ is not convex, and we do not adopt this acceleration anymore.
>
> In line 6 of Algorithm 2, we do not assume "that the optimal solutions for are equal to each other."
> This is a step of the PAV algorithm. In this step, we enforce to solve the problem $\min_z \sum_{i=s_k}^{s_{k+t+1}}\theta_i(z)$, where the optimal solution is $v_{[s_k,s_{k+t+1}]}$. This step is called  "merge the consecutive out-of-order blocks."
>
> When $\theta_i$ are all convex functions, the PAV algorithm returns a global minimum, which is proved in [3]. For human risk minimization, the PAV algorithm finds a point that satisfies the first-order condition, which is proved in [Theorem 3, 4].
>
> **Statement about assumptions**
>
> Sorry that the statement that Assumptions 4 and 5 are weaker than Assumption 2 may not be accurate. It is preferable to state that Assumptions 4 and 5 are more practical than Assumption 2, as they are easier to verify in practice. As mentioned in line 226-234, the full row rank property of the data matrix is often assumed in the high dimensional setting classification and Assumption 5 is satisfied by weakly convex functions that are Lipschitz continuous.
>
> **Specific information on experiments**
>
> Sure. We will present the formulations in Section 5.2 in the manuscript.
> In Section 5.2, we used logistic loss and $\ell_2$ norm regularizer. In Section 5.3, we used both logistic loss and hinge loss and $\ell_2$ norm regularizers as in [5]. Detailed information on loss functions, regularizers, and other settings can be found in Appendix B.5.
>
> **Typos and Minore issues**
>
> Thank you for pointing out our typos and suggestions in the Minor issue. We will make the necessary corrections in the subsequent manuscript. Here we provide a detailed explanation of two points.
>
> For Minor issue 4, fairness means that our predictions remain consistent across different groups. For example, in the examples used in the experimental section, we examine whether there are differences in predictions between different races when predicting gender. The fairness metrics are detailed in Appendix B.5.
>
> For Minor issue 5, since we set $D=-\text{diag}(\boldsymbol{y})X$ in line 125, the expression $\boldsymbol{z}=D\boldsymbol{w}$ in line 127 is correct.
>
> **References**
>
> [1] Mehta, R., Roulet, V., Pillutla, K., Liu, L., & Harchaoui, Z. (2023, April). Stochastic Optimization for Spectral Risk Measures. In International Conference on Artificial Intelligence and Statistics (pp. 10112-10159). PMLR.
>
> [2] Hoheisel, T., Laborde, M., & Oberman, A. On proximal point-type algorithms for weakly convex functions and their connection to the backward Euler method. Optimization Online.
>
> [3] Best, M. J., Chakravarti, N., & Ubhaya, V. A. (2000). Minimizing separable convex functions subject to simple chain constraints. SIAM Journal on Optimization, 10(3), 658-672.
>
> [4] Cui, X., Jiang, R., Shi, Y., and Yan, Y. (2023). Decision making under cumulative prospect theory: An alternating direction method of multipliers. arXiv preprint arXiv:2210.02626.
>
> [5] Hu, S., Ying, Y., Lyu, S., et al. (2020). Learning by minimizing the sum of ranked range. Advances in Neural Information Processing Systems, 33:21013–21023.

---

### Official Review · Reviewer_Gyag · 2023-07-06

**Soundness:** 3 good
**Presentation:** 2 fair
**Contribution:** 3 good
**Rating:** 6
**Confidence:** 1

**Summary:**

This paper proposes a unified framework for rank-based loss minimization based on the ADMM algorithm. The paper proposes to apply a pool adjacent violators (PAV) algorithm to solve one of the subproblems of ADMM. Numerical experiments show that the proposed algorithm outperforms the existing ones.

**Strengths:**

+ The problem of rank-based loss minimization is very important.
+ The proposed PAV algorithm looks interesting.
+ Convergence of the algorithm is theoretically analyzed.

**Weaknesses:**

- It seems that Eq.(1) assumes a linear model and it is unclear how to generalize to non-linear settings.

**Questions:**

- Can the proposed method be applied to nonlinear models?

**Limitations:**

Limitations are not discussed in the paper.

---

> ### Author Rebuttal · Authors · 2023-08-04
>
> We thank the reviewer Gyag for the feedback and comments!
>
> Eq. (1) indeed assumes a linear model. One reason for making this assumption is that currently, many studies or experimental parts related to rank-based loss predominantly concentrate on linear models [1-3]. So far, there has been limited research on ADMM with nonlinear constraints. However, as stated below, our method can be applied to non-linear models.
>
> **Applying the method to a non-linear model**
>
> The linear model in Eq. (1) can be replaced with a non-linear model. The rank-based loss in Eq. (1) can be rewritten as follows:
> \begin{equation}
> \Omega(\boldsymbol{w}):=\sum_{i=1}^{n} \sigma_i \boldsymbol{l}_{[i]}\left(- \boldsymbol{y} \odot (h(\boldsymbol{w}))\right)
> \end{equation}
> where $h(\boldsymbol{w}):\mathbb{R}^d\to\mathbb{R}^n$ is a differentiable non-linear function.
>
> Let us explain how to adapt the ADMM algorithm. Now the constraint in eq. (9)  should be $\boldsymbol{z} = Dh(\boldsymbol{w})$, where $D = -\text{diag}(\boldsymbol{y})$. Information about the samples $X$ is encompassed in $h(\boldsymbol{w})$, but to maintain consistency in notation, we still retain $D$. The $z$-subproblem is the same as the linear model version.
>
> In order to keep our theory still valid, we need to make modifications in the following two aspects:
>
> **1. Solving $w$-subproblem**
>
> Due to the inclusion of the non-linear term $h(\boldsymbol{w})$, the $\boldsymbol{w}$-subproblem may no longer be strongly convex, leading to potential inability to exactly solve the $\boldsymbol{w}$-subproblem. To overcome this limitation, we assume $\boldsymbol{w}$-subproblem is solved inexactly such that $\text{dist}\left(0,\partial_{\boldsymbol{w}} L\_\rho\left(\boldsymbol{z}^{k+1},\boldsymbol{w}^{k+1},\boldsymbol{\lambda}^{k}\right)+r(\boldsymbol{w}^{k+1}-\boldsymbol{w}^{k})\right)\leq O(\epsilon)$, where $\epsilon$ is the accuracy in Theorems 1, 2.  Moreover, we assume the solution to the $\boldsymbol{w}$-subproblem is an $\epsilon_k$ optimal solution ($\epsilon_k\geq 0$) and $\sum\_{k=1}^{\infty}\epsilon\_k<\infty$. We say $\hat{x}$ is an $\tilde{\epsilon}$ optimal solution for $\min_x f(x)$ if $f(\hat{x})\leq \min_x f(x) +\tilde{\epsilon}$.
>
> In this way, we can obtain the descent of $\boldsymbol{w}$-subproblem:
> $$
> L_\rho(\boldsymbol{\boldsymbol{z}}^{k+1},{\boldsymbol{w}}^k,\boldsymbol{\lambda}^k)\geq L_\rho(\boldsymbol{\boldsymbol{z}}^{k+1},{\boldsymbol{w}}^{k+1},\boldsymbol{\lambda}^k) +\frac{r}{2}\\|{\boldsymbol{w}}^{k+1}-{\boldsymbol{w}}^k\\|^2-\epsilon_k.
> $$
>
> **2. Convergence guarantee of our algorithm**
>
> We need the following additional assumptions to guarantee the convergence of the algorithm:
> - $h(\boldsymbol{w})$ and $\nabla h(\boldsymbol{w})$ are Lipschitz continuous in any sublevel sets.
> - $\boldsymbol{\lambda}^k \in Im(D\nabla h(\boldsymbol{w}^k))~ \forall k$.
>
> The first controls $\\|\boldsymbol{w}^{k+1}-\boldsymbol{w}^k\\|$ by $\\|h(\boldsymbol{w}^{k+1})-h(\boldsymbol{w}^{k})\\|$ or $\\|\nabla h(\boldsymbol{w}^{k+1})-\nabla h(\boldsymbol{w}^{k})\\|$. The latter is introduced to replace Assumption 4 in our paper.
>
> As a result, our main findings can be rewritten as follows:
>
> **Theorem 1 (modified)**   Under the same assumptions and settings as in Theorem 1, along with the two additional assumptions mentioned above,  Algorithm 1 can find an $\epsilon$-KKT point $(\boldsymbol{\boldsymbol{z}}^{k+1},{\boldsymbol{w}}^{k+1},\boldsymbol{\lambda}^{k+1})$ within $O(1/\epsilon^2)$ iterations, i.e.,
>       \begin{equation}
>         \text{dist}\left(-\boldsymbol{\lambda}^{k+1},\partial\Omega\left(\boldsymbol{\boldsymbol{z}}^{k+1}\right)\right)\leq \epsilon,\quad
>                    \text{dist}\left(\left(D\nabla h(\boldsymbol{w}^{k+1})\right)^T{\boldsymbol{\lambda}}^{k+1},\partial g\left(\boldsymbol{w}^{k+1}\right)\right)\leq O(\epsilon),\quad
>         \\|\boldsymbol{\boldsymbol{z}}^{k+1}-D{\boldsymbol{w}}^{k+1}\\|\leq \epsilon.
>     \end{equation}
>
> **Theorem 2 (modified)** Under the same assumptions and settings as in Theorem 2, along with the two additional assumptions mentioned above (Assumption 4 is no longer required),  Algorithm 1 can find an $\epsilon$-KKT point $(\boldsymbol{\boldsymbol{z}}^{k+1}, {\tilde{\boldsymbol{w}}^{k+1}},\boldsymbol{\lambda}^{k+1})$ within $O(1/\epsilon^4)$ iterations, i.e.,
>
> \begin{equation}
>         \text{dist}\left(-\boldsymbol{\lambda}^{k+1},\partial\Omega\left(\boldsymbol{\boldsymbol{z}}^{k+1}\right)\right)\leq \epsilon,\quad
>         \text{dist}\left(\left(D\nabla h(\tilde{\boldsymbol{w}}^{k+1})\right)^T{\boldsymbol{\lambda}}^{k+1},\partial g\left(\tilde{\boldsymbol{w}}^{k+1}\right)\right) \leq O(\epsilon),\quad
>         \\|\boldsymbol{\boldsymbol{z}}^{k+1}-D\tilde {\boldsymbol{w}}^{k+1}\\|\leq \epsilon.
>     \end{equation}
>
> **The potential practical limitations of the proposed method**
>
> To ensure the effective utilization of our algorithm, it is crucial for the individual loss to demonstrate monotonicity. This property enables the application of the PAV algorithm for solving the $z$-subproblems. Some examples of loss functions that satisfy this criterion include logistic loss, hinge loss, and exponential loss. On the other hand, when confronted with a substantially large sample size, the computational efficiency of the PAV algorithm could be impacted, thereby potentially constraining the overall effectiveness of the entire algorithm. We will point out the limitations in our paper.
>
> **References**
>
> [1] Mehta, R., Roulet, V., Pillutla, K., Liu, L., & Harchaoui, Z. (2023, April). Stochastic Optimization for Spectral Risk Measures. In International Conference on Artificial Intelligence and Statistics (pp. 10112-10159). PMLR.
>
> [2] Hu, S., Ying, Y., & Lyu, S. (2020). Learning by minimizing the sum of ranked range. Advances in Neural Information Processing Systems, 33, 21013-21023.
>
> [3] Leqi, L., Prasad, A., & Ravikumar, P. K. (2019). On human-aligned risk minimization. Advances in Neural Information Processing Systems, 32.

---

> > ### Comment · Reviewer_Gyag · 2023-08-18
> > **Thank you for the response**
> >
> > Thank the authors for the response. The nonlinear extension looks interesting and with this incorporated, the paper overall looks solid to me. I am willing to raise the rating to 6.

---

### Author Rebuttal · Authors · 2023-08-05

We truly thank all reviewers’ insightful and constructive suggestions, which helped to significantly improve our paper!

**The more experiments about Figure 1**

To explain the phenomenon observed in Figure 1(h), we conducted experiments with increased sample sizes. The corresponding results are included in the PDF file.

---

### Decision · Program_Chairs · 2023-09-21

**Decision:**

Accept (poster)

**Comment:**

The paper proposes ADMM style algorithms for rank based losses such as spectral risks. The main contribution is to devise a new algorithm through Pool Adjacent violators.